EMBO
Molecular Medicine

# Nf1 loss promotes Kras-driven lung adenocarcinoma and results in Psat1-mediated glutamate dependence

Xiaojing Wang[1,*,†] (ID), Shengping Min[1,†], Hongli Liu[2], Nan Wu[1], Xincheng Liu[1], Tao Wang[1], Wei Li[1], Yuanbing Shen[1], Hongtao Wang[3], Zhongqing Qian[3], Huanbai Xu[4], Chengling Zhao[1] & Yuqing Chen[1,**] (ID)

## Abstract

Mutations to *KRAS* are recurrent in lung adenocarcinomas (LUAD) and are daunting to treat due to the difficulties in KRAS oncoprotein inhibition. A possible resolution to this problem may lie with co-mutations to other genes that also occur in *KRAS*-driven LUAD that may provide alternative therapeutic vulnerabilities. Approximately 3% of *KRAS*-mutant LUADs carry functional mutations in *NF1* gene encoding neurofibromin-1, a negative regulator of focal adhesion kinase 1 (FAK1). We evaluated the impact of *Nf1* loss on LUAD development using a CRISPR/Cas9 platform in a murine model of *Kras*-mutant LUAD. We discovered that *Nf1* deactivation is associated with Fak1 hyperactivation and phosphoserine aminotransferase 1 (Psat1) upregulation in mice. *Nf1* loss also accelerates murine *Kras*-driven LUAD tumorigenesis. Analysis of the transcriptome and metabolome reveals that LUAD cells with mutation to *Nf1* are addicted to glutamine metabolism. We also reveal that this metabolic vulnerability can be leveraged as a treatment option by pharmacologically inhibiting glutaminase and/or Psat1. Lastly, the findings advocate that tumor stratification by co-mutations to *KRAS/NF1* highlights the LAUD patient population expected to be susceptible to inhibiting PSAT1.

**Keywords** glutaminolysis; KRAS; lung cancer; NF1; PSAT1
**Subject Categories** Cancer; Metabolism; Respiratory System

## Introduction

Lung cancer remains the leading cause of cancer death globally, accounting for about 20% (1.6 million) of the total cancer deaths annually (Aggarwal *et al*, 2016; Hirsch *et al*, 2017). With a very high annual mortality-to-incidence ratio of 0.89 (Mao *et al*, 2016; Hirsch *et al*, 2017), the economic costs directly related to lung cancer are staggering. In the United States, annual lung cancer expenditures are approximately $13.1 billion and lost productivity due to premature lung cancer-related deaths is an estimated additional $36.1 billion (Mariotto *et al*, 2011; Howlader *et al*, 2015). In the E.U., lung cancer accounts for ~15% of the overall cancer costs (€18.8 billion; Luengo-Fernandez *et al*, 2013).

Approximately 40% of lung cancers are adenocarcinomas (LUADs), usually originating from the peripheral lung tissue (Coroller *et al*, 2015). As can be expected, smoking has the strongest association to LUAD onset and prognosis (Network, 2014); nonetheless, LUAD is also the most common form of lung cancer in non-smokers (Gharibvand *et al*, 2017). Through a concerted effort of several multiplatform genomic profiling studies, the Kirsten rat sarcoma viral oncogene homolog gene (*KRAS*) was found to account for 25% of the molecular aberrations identified in various driver oncogenes in LUAD tumors (Tsao *et al*, 2016). However, *KRAS*-based molecular-targeted therapies have largely been unsuccessful, primarily due to the difficulties associated with directly inhibiting oncogenic KRAS (Downward, 2015). *KRAS*-mutant LUADs with distinct immune profiles and therapeutic susceptibilities have been distinguished based on co-occurring genomic alterations (Skoulidis *et al*, 2015). As such, characterizing and targeting other functionally relevant molecular aberrations in *KRAS*-mutant LUADs can be used as an alternative approach to managing these LUADs (Romero *et al*, 2017).

1 Anhui Clinical and Preclinical Key Laboratory of Respiratory Disease, Department of Respiration, First Affiliated Hospital, Bengbu Medical College, Bengbu, Anhui Province, China
2 Department of Gynecological Oncology, First Affiliated Hospital, Bengbu Medical College, Bengbu, Anhui Province, China
3 Department of Immunology, Bengbu Medical College, Bengbu, Anhui Province, China
4 Department of Endocrinology and Metabolism, Shanghai Jiaotong University Affiliated First People's Hospital, Shanghai, China
*Corresponding author. Tel: +86 15105528215; Fax: +86 05523070260; E-mail: wangxiaojing8888@163.com
**Corresponding author. Tel: +86 13695528585; Fax: +86 05523070260; E-mail: bbmccyq@126.com
†These authors contributed equally to this work

To that end, putative loss-of-function mutations in the tumor suppressor *NF1* (neurofibromin-1) have been identified in about 11% of all LUAD tumors and occur in approximately 3% of *KRAS*-mutant LUAD tumors (Network, 2014), suggesting that *NF1* loss of function may play an important role in a subset of *KRAS*-mutant LUAD tumors. The *NF1* gene encodes a GTPase-activating protein that regulates GTP-bound RAS' GTPase activity, thereby functioning as an "off" signal for RAS GTPase (Ratner & Miller, 2015). Thus, *NF1* loss promotes the activity of RAS effector pathways with prominent roles in oncogenesis, such as the RAS–MAPK pathway (Ratner & Miller, 2015). Interestingly, the NF1 protein has been proven to co-localize and interact with the tyrosine kinase focal adhesion kinase 1 (FAK1) in mammalian cells (Kweh *et al*, 2009). This is noteworthy, as FAK1 is upregulated in > 80% of solid tumors and serves as a protein scaffold for several important oncogenic binding partners involved in cancer cell survival, proliferation, invasiveness, and angiogenesis (Lenzo & Cance, 2017). Of relevance here, previous research has revealed that Fak1 activity is necessary for tumor progression in *Kras;Cdkn2a*-mutant and *Kras;Lkb1*-mutant murine models of LUAD (Konstantinidou *et al*, 2013; Gilbert-Ross *et al*, 2017), suggesting that Fak1 may be a critical oncogene in certain subsets of *KRAS*-mutant LUAD tumors.

On this basis, we chose to target the Nf1-Fak1 axis in the *loxP*-STOP-*loxP* (LSL)-*Kras*^G12D/+; *Tp53*^flox/flox (*p53*) (referred to as KP) genetically engineered mouse model (GEMM) of human LUAD by employing a CRISPR/Cas9 platform (Romero *et al*, 2017). We show that loss of *Nf1* is associated with Fak1 hyperactivation and upregulation of the glutamine-metabolizing enzyme phosphoserine aminotransferase 1 (Psat1) in mice. We also found that loss of *Nf1* accelerates murine *Kras*-dependent LUAD tumorigenesis. Using analysis of the transcriptome and metabolome, we also demonstrate that tumors with mutation to *Nf1* are reliant upon α-ketoglutarate (α-KG) production from glutamate via the glutaminase–Psat1 pathway. We also demonstrate that this metabolic vulnerability can be

leveraged as a treatment strategy by pharmacologically inhibiting glutaminase and/or Psat1. Lastly, the work suggests that tumor stratification by co-mutations to *KRAS/NF1* highlights the LAUD patient population expected to benefit from inhibiting PSAT1.

# Results

## *Nf1* loss accelerates *Kras;p53*-mutant LUAD tumorigenesis

As *TP53* mutations show prognostic significance in *KRAS*-mutant LUAD cases and *NF1*-mutant LUAD tumors show significantly higher rates of *TP53* mutation relative to *KRAS*-mutant LUAD tumors (Redig *et al*, 2016), we investigated the cooperative effect of silencing Nf1 and p53 expression on the oncogenicity of constitutively active *Kras* mutants. Using CRISPR-Cas9 technology, the *Kras*^LSL-G12D/+; *p53*^fl/fl (KP) murine model of *Kras*-driven LUAD was engineered to silence Nf1 and p53 expression (Fig 1A). Twenty weeks after intratracheal infection of KP mice with lentiviral vectors expressing sgRNAs against *Nf1* (sgNf1.1, sgNf1.2, and sgNf1.3) or *tdTomato* (sgTom) as control, we observed significant increases in average and total tumor volume in KP mice infected with sgRNAs against *Nf1* as compared to sgTom (Fig 1B and C). We also observed significant increases in tumor burden 21 weeks after infection with sgRNAs against *Nf1* as compared to sgTom (Fig 1D). Histological grading analysis also revealed an overrepresentation of grade 3 and 4 tumors in sgNf1-infected KP mice compared to sgTom-infected KP mice (Fig 1E). Notably, the highest proportion of grade 4 tumors (which were completely absent in sgTom-infected KP mice) was observed in KP mice infected with sgNf1.3. Accordingly, the mitotic indices (as observed on pHH3-stained slides) were significantly greater for sgNf1-infected KP mice relative to sgTom-infected KP mice (Fig 1F). Comparative qPCR and immunohistochemical analysis of LUAD tumor sections 21 weeks after infection

**Figure 1. Nf1 loss activates Fak1 and accelerates murine LUAD tumorigenesis.**

A   Schematic of *Kras*^LSL-G12D/+; *p53*^fl/fl (KP) mice intratracheally infected with pSECC lentiviruses containing sgNf1 or control sgTom. Mouse tumor burden was tracked by micro-computed tomography (micro-CT) for 5 months post-infection, and lungs were finally harvested 21 weeks post-infection. Tumors were dissected out for immunohistochemical (IHC) staining and generation of the tumor-derived parental KP cell line.

B   Quantification by micro-CT of mean tumor volumes in KP mice derived from 50 randomly selected tumors at 8, 12, 16, and 20 weeks post-infection with control sgTom, sgNf1.1, sgNf1.2, or sgNf1.3 (*n* = 18 mice, 21 mice, 21 mice, and 20 mice, respectively).

C   Quantification by micro-CT of total tumor volume per mouse in KP mice after infection with control sgTom, sgNf1.1, sgNf1.2, or sgNf1.3 at 8, 12, 16, and 20 weeks post-infection (*n* = 18 mice, 21 mice, 21 mice, and 20 mice, respectively).

D   Depictions and quantitation of total tumor burden (total tumor area/total lung area) in KP mice after infection with control sgTom, sgNf1.1, sgNf1.2, or sgNf1.3 at 21 weeks after infection (*n* = 18 mice, 21 mice, 21 mice, and 20 mice, respectively).

E   Distribution of histological tumor grades in KP mice after infection with control sgTom, sgNf1.1, sgNf1.2, or sgNf1.3 (*n* = 50 tumors each) at 21 weeks after infection.

F   Assessment of the mitotic index of tumor cells by phosphorylated-histone H3 (pHH3)-positive nuclei density in KP mice LUAD tumors at 21 weeks after infection with control sgTom, sgNf1.1, sgNf1.2, or sgNf1.3 (*n* = 50 tumors each).

G   Nf1 mRNA expression in LUAD tumor sections 21 weeks after infection with control sgTom, sgNf1.1, sgNf1.2, or sgNf1.3 (*n* = 50 tumors each).

H   Representative hematoxylin and eosin (H&E) and p-Fak1 immunohistochemical (IHC) staining of LUAD tumor sections 21 weeks after infection with control sgTom (grade 1 depicted), sgNf1.1 (grade 3 depicted), sgNf1.2 (grade 3 depicted), or sgNf1.3 (grade 3 depicted) (*n* = 50 tumors each). H&E scale bars (low-magnification top row = 100 μm, high-magnification bottom row = 250 μm); p-Fak1 IHC scale bars (low-magnification top row = 250 μm, high-magnification bottom row = 500 μm).

I   Quantification of p-Fak1 IHC signals in LUAD tumor sections 21 weeks after infection with control sgTom, sgNf1.1, sgNf1.2, or sgNf1.3 (*n* = 50 tumors each).

J   Quantification of p-Fak1 IHC signals in sgNf1.3 LUAD tumor sections analyzed by tumor grade (*n* = 50 tumors).

K   Quantification of Psat1 mRNA expression in LUAD tumor sections 21 weeks after infection with control sgTom, sgNf1.1, sgNf1.2, or sgNf1.3 (*n* = 50 tumors each).

L   Quantification of Psat1 mRNA expression in sgNf1.3 LUAD tumor sections analyzed by tumor grade (*n* = 50 tumors).

Data information: *P*-values are reported in Appendix Table S3. In bar charts and line graphs, data presented as means with error bars representing standard deviations (SDs). For boxplots, whiskers indicate the minimum and maximum values, the upper and lower perimeters represent the first and third quartiles, the midline represents the median value, and the x symbol represents the mean.

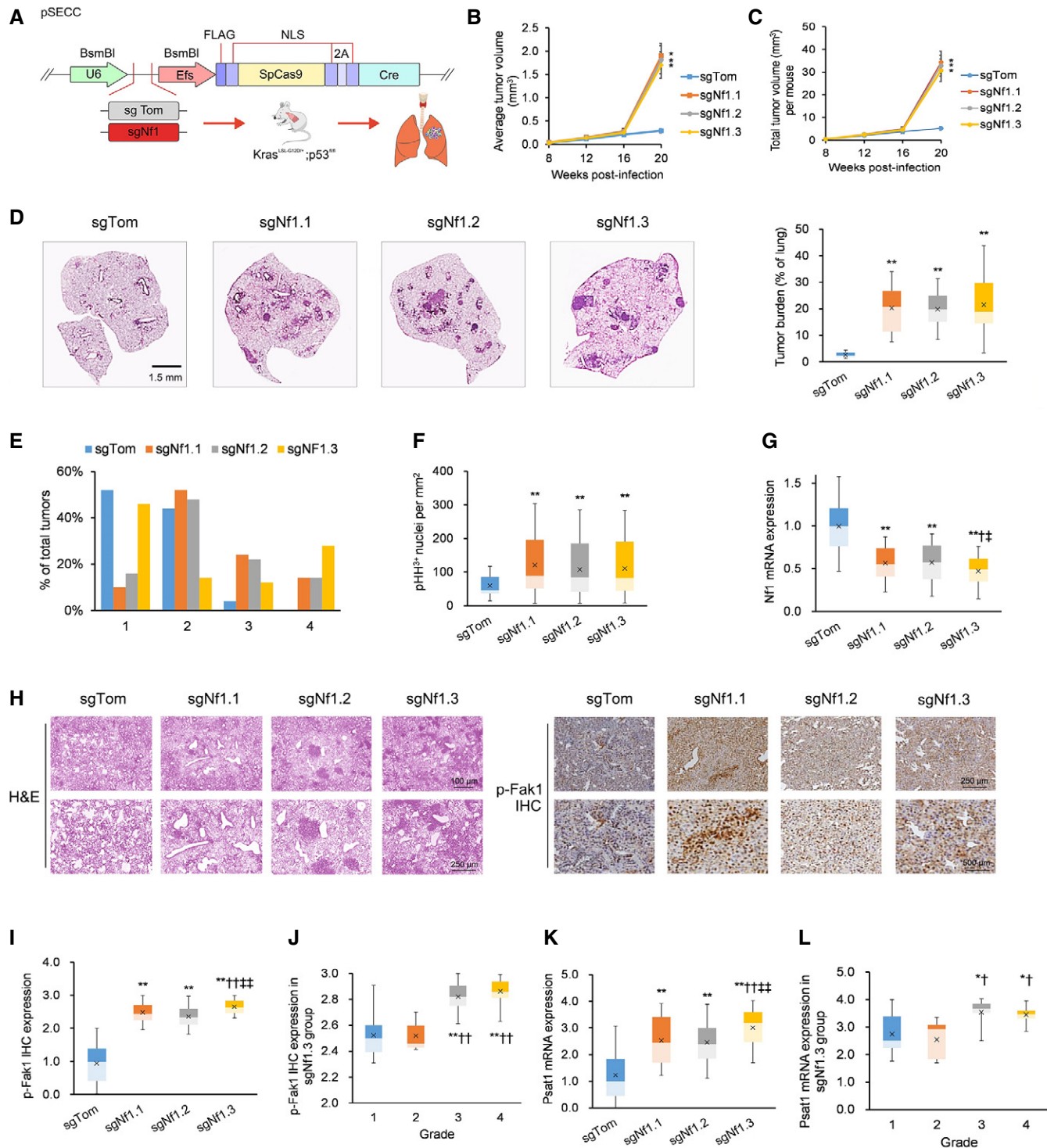

**Figure 1.**

of KP mice confirmed significant Nf1 mRNA downregulation (Fig 1G), p-Fak1 protein upregulation (Fig 1H–J), and Psat1 mRNA upregulation (Fig 1K and L) in sgNf1-infected mice, with the most profound effects observed in the sgNf1.3-infected group. On the basis of this combined evidence, we selected sgNf1.3 as the sgRNA for Nf1 silencing in all further *in vitro* and *in vivo* experiments.

Using CRISPR-Cas9 technology, murine $Kras^{\text{LSL-G12D/}+}$; $p53^{\text{fl/fl}}$ (KP) cells were engineered to silence *Nf1* using sgNf1.3 (KPΔNF1) and *Fak1* using sgFak1.1 (KPΔFAK1; Appendix Fig S1A and B). We generated subcutaneous and orthotopic transplants of non-recombinant KP and recombinant KPΔNF1 cells to answer whether *Nf1* inactivation confers a selective growth advantage *in vivo*. From day 10

post-implantation into nude mice, we observed a consistent increase in subcutaneous tumor volumes for mice with KPΔNF1 tumors (Appendix Fig S1C and D) as well as an ~4-fold increase in subcutaneous tumor masses for mice with KPΔNF1 tumors (Appendix Fig S1E and F). Ki-67, a cellular proliferation marker, was significantly more enriched in nuclei from subcutaneous KPΔNF1 tumors (Appendix Fig S1G and H). We also observed a consistent increase in tumor growth for mice with orthotopic KPΔNF1 tumors (Appendix Fig S1I and J). *In vitro*, cumulative population doublings were found to progressively increase in KPΔNF1 cells (Appendix Fig S1K).

## *Nf1* loss also accelerates *Kras*-mutant;*p53*-WT murine LUAD tumorigenesis

The previous experiments demonstrated that *Nf1* loss accelerates *Kras*;*p53*-mutant LUAD tumorigenesis. However, the effects of *Nf1* loss on *Kras*-mutant;*p53*-WT LUAD tumorigenesis remain unknown. Therefore, we investigated p53's possible involvement in *NF1*-mediated KRAS-LUAD. p53 protein expression was pharmacologically induced in the patient-derived *KRAS*-mutant/*NF1*-mutant/*TP53*-WT LUAD cell lines PDKN1 and PDKN2, the human *KRAS*-mutant/*NF1*-WT/*TP53*-WT LUAD cell line SW1573, and the murine *Kras*-mutant/*Nf1*-WT/*p53*-WT LUAD clones LKR10 and LKR13 by incubating them with a DNA intercalator doxorubicin for 6 h (Fig 2A, Appendix Fig S2A). Western blot analysis validated NF1 upregulation and downregulated FAK1 activation following transduction of doxorubicin-treated PDKN1 and PDKN2 cell lines with an *NF1*-expression vector (Fig 2B and C; Appendix Fig S2B and C). Moreover, we validated Nf1 downregulation and Fak1 activation following transduction of doxorubicin-treated SW1573, LKR10, and LKR13 clones with sgNf1.3 (Fig 2D–F; Appendix Fig S2D–F).

We examined the effects of *Nf1* overexpression and loss across multiple *p53*-WT models. First, overexpression of NF1 in the doxorubicin-treated PDKN1 cell line led to repression of FAK1 activity as evidenced by downregulation of the FAK1 target genes *PSAT1*, *AREG*, and *PEG10* (Golubovskaya *et al*, 2009; Fig 2G). Moreover,

overexpression of *NF1* decreased subcutaneous tumor volumes for doxorubicin-treated PDKN1 and PDKN2 cell lines (Fig 2H and I). Second, silencing NF1/Nf1 in doxorubicin-treated SW1573 cells as well as doxorubicin-treated LKR10 and LKR13 clones resulted in significantly greater subcutaneous tumor volumes across all three models (Fig 2J–L). Third, we examined the effects of Nf1 silencing in $Kras^{G12D/+}$; $p53^{+/+}$ (K-only) autochthonous tumors. sgNf1.3 mice displayed a significantly greater tumor burden (Fig 2M), an overrepresentation of grade 2 and 3 tumors (Fig 2N), and nuclear Ki-67 enrichment in their K-only autochthonous tumors (Fig 2O) relative to sgTom mice. Note that these grading results for K-only tumors are lower than those observed in the $Kras^{LSL-G12D/+}$; $p53^{fl/fl}$ (KP) tumors from sgNf1.3 mice (Fig 1D), which can likely be attributed to the tumor-suppressive effects of p53 expression in the K-only autochthonous tumors. To summarize, *Nf1* loss exacerbates *Kras*-driven lung adenocarcinogenesis even in the background of WT p53 expression.

### *Nf1* loss enhances Fak1 activation *in vitro*

We analyzed Fak1 activation in engineered KPΔNF1 and KPΔFAK1 cells. Immunoblot and qRT-PCR analyses revealed that Fak1, phosphorylated Fak1 (p-Fak1), and its target genes (*Psat1*, *Areg*, and *Peg10*; Golubovskaya *et al*, 2009) were significantly upregulated in KPΔNF1 cell lines (Appendix Fig S3A and B). Because phosphatidylinositol -4,5-bisphosphate (PIP2) is a Fak1 activator, recombinant and non-recombinant KP cells were incubated in 10 μM PIP2 for 6 h. We observed that upregulation of p-Fak1 [Appendix Fig S3C; with concomitant upregulation of its target genes *Psat1*, *Areg*, and *Peg10* (Appendix Fig S3D–F; Golubovskaya *et al*, 2009)] only occurs in PIP2-stimulated parental KP cells and is further upregulated in PIP2-stimulated KPΔNF1 clones.

### *Nf1* loss enhances *Kras*-mutant fermentation and glutamine dependence

Having better established the role of NF1 loss in KRAS-dependent LUAD, discovery of metabolic susceptibilities in *Kras*;*Nf1*-mutant

---

**Figure 2. Nf1 loss accelerates tumor growth in p53-independent manner.**

A   Western blotting analysis of p53 expression in various *p53/TP53*-WT cell lines incubated with the DNA-intercalating agent doxorubicin (DOXO, 0.2 μg/ml for 6 h) to induce p53 stabilization (*n* = 3 biological replicates). Full experimental data provided in Appendix Fig S2A.

B, C   Western blotting analysis confirming NF1 upregulation and p-FAK1 downregulation in (B) DOXO-treated PDKN1 cells and (C) DOXO-treated PDKN2 cells transduced with PGK-NF1 (*n* = 3 biological replicates). Full experimental data provided in Appendix Fig S2B and C.

D   Western blotting analysis confirming NF1 downregulation and p-FAK1 upregulation in DOXO-treated SW1573 cells transduced with sgNf1.3 (*n* = 3 biological replicates). Full experimental data provided in Appendix Fig S2D.

E, F   Western blotting analysis confirming Nf1 downregulation and p-Fak1 upregulation in (E) DOXO-treated LKR10 and (F) DOXO-treated LKR13 clones transduced with sgNf1.3 (*n* = 3 biological replicates). Full experimental data provided in Appendix Fig S2E and F.

G   FAK1 target gene expression in DOXO-treated PDKN1 cells transduced with PGK-control or PGK-NF1 (*n* = 3 biological replicates).

H, I   Subcutaneous tumor volumes of DOXO-treated PDKN1 and PDKN2 cells transduced with PGK-control or PGK-NF1 (*n* = 18 each).

J–L   Subcutaneous tumor volumes of (J) SW1573, (K) LKR10, and (L) LKR13 cells transfected with sgTom or sgNf1.3 (*n* = 18 each).

M   Depictions and quantitation of $Kras^{G12D/+}$; $p53^{+/+}$ (K-only) autochthonous tumor burden (total tumor area/total lung area) in pSECC-sgTom (*n* = 18 mice) or pSECC-sgNf1.3 mice (*n* = 21 mice).

N   Analysis of tumor grades in K-only autochthonous tumors derived from pSECC-sgTom (*n* = 18 mice) or pSECC-sgNf1.3 mice (*n* = 21 mice).

O   Quantification of Ki-67-positive nuclei per mm$^2$ in K-only autochthonous tumors derived from pSECC-sgTom (*n* = 18 mice) or pSECC-sgNf1.3 mice (*n* = 21 mice).

Data information: *P*-values are reported in Appendix Table S3. In bar charts and line graphs, data presented as means with error bars representing standard deviations (SDs). For boxplots, whiskers indicate the minimum and maximum values, the upper and lower perimeters represent the first and third quartiles, the midline represents the median value, and the x symbol represents the mean.
Source data are available online for this figure.

---

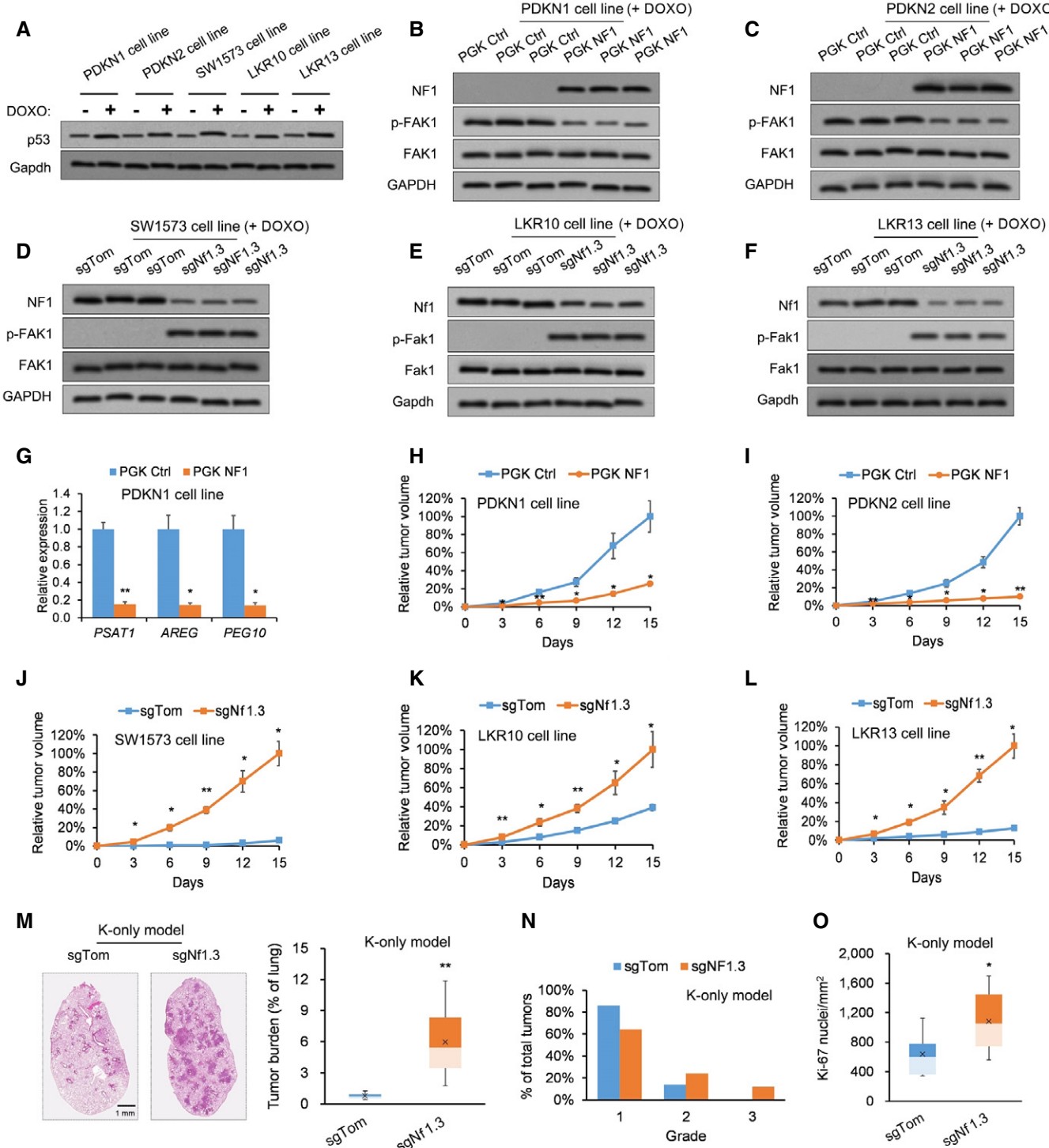

Figure 2.

LUAD cells that can be therapeutically exploited was pursued next. To answer whether these cells exhibit the marked glutamine dependence typical of many cancer cells, we assessed metabolic changes associated with glutamine restriction/depletion *in vitro*. Analyses of glucose/lactate levels (Fig 3A) and glutamine levels (Fig 3B) reveal that both the fermentation and the utilization of

glutamine are significantly enhanced in KPΔNF1 cells. Incidentally, inhibition of glycolysis (by incubating cells in 2DG) was mildly cytotoxic to *Kras*-mutant cells but extremely cytotoxic to *Kras;Nf1*-mutant cells (Appendix Fig S4A). An isotopomer of glucose in which every carbon was $^{13}$C-labeled (referred to as [U$^{13}$C]) was used as a metabolic tracer in KPΔNF1 cells. They

showed a lowered input into the Krebs cycle and its intermediates of $^{13}C$ derived from [U$^{13}$C] glucose (Appendix Fig S4B–D). Restriction of glutamine uptake by treatment with γ-l-glutamyl-p-nitroanilide (GPNA, a small-molecule inhibitor of the glutamine transporter SLC1A5 that is expressed in 74% of LUADs; Hassanein *et al*, 2013; Fig 3C) or decreased glutamine supplementation (Fig 3D) significantly decreased KPΔNF1 cell viability and proliferation, respectively. Taken together, KPΔNF1 cells display enhanced fermentation and heightened glutamine dependence.

### Nf1 loss enhances expression of the glutamine-metabolizing enzyme Psat1

We performed a transcriptomic microarray screen to identify changes in the mRNA expression of genes in response to *NF1* silencing. Transcriptomic screening of patient-derived *NF1*-mutant PDKN1 and PDKN2 cells passaged for 14 population doublings identified the glutamine-metabolizing enzyme *PSAT1* among the genes with the most profound changes in expression relative to patient-derived *NF1*-WT PDK cells (Fig 3E). As previously published bioinformatics

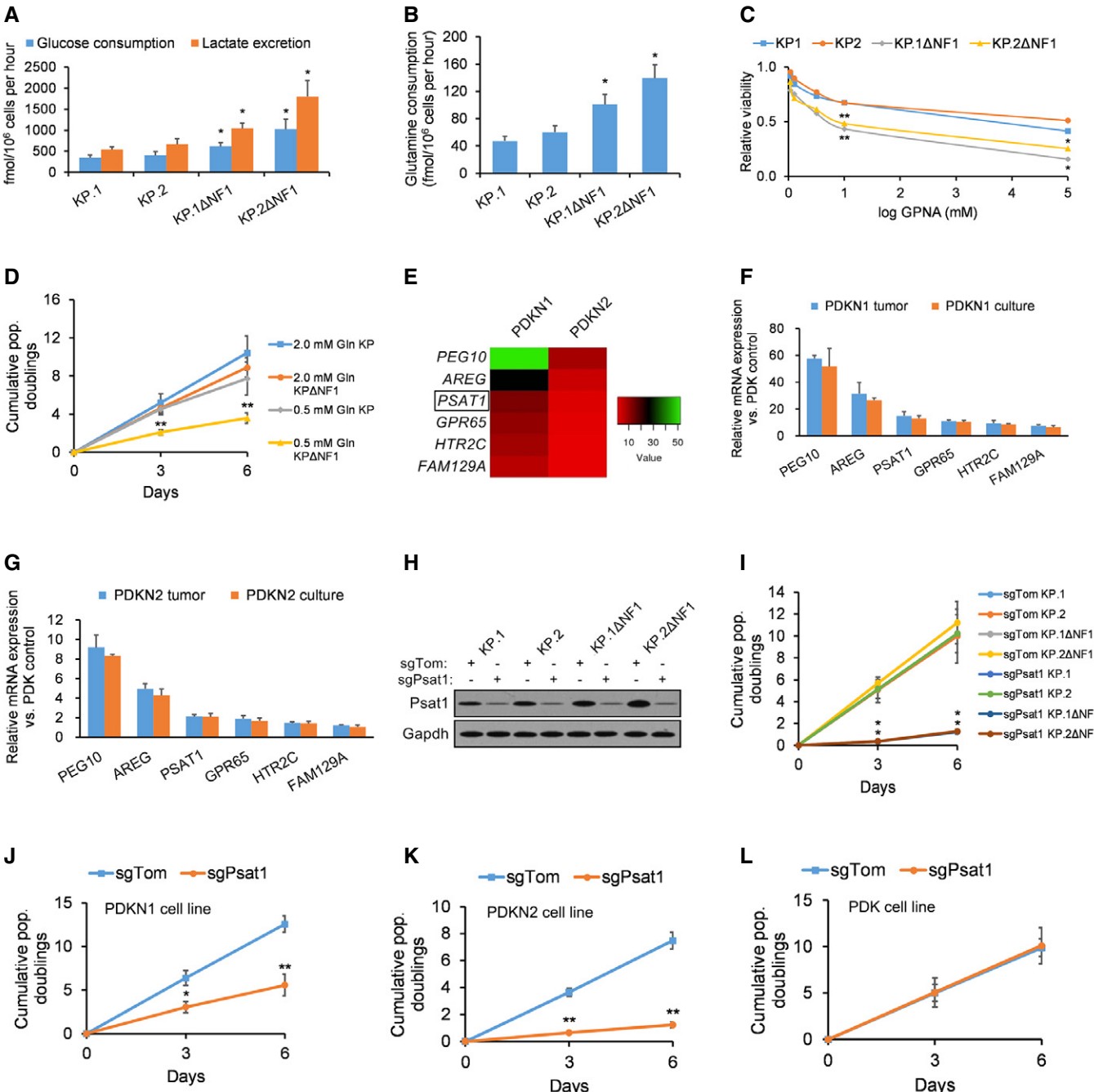

**Figure 3.**

**Figure 3. Nf1-silenced cells are sensitive to reduced glutamine levels.**

A    Glucose consumption and lactate excretion in KP and KPΔNF1 clones normalized by cell count ($n$ = 4 biological replicates).

B    Glutamine consumption in KP and KPΔNF1 clones normalized by cell count ($n$ = 4 biological replicates).

C    Relative viability of KP and KPΔNF1 cells after 72 h of GPNA treatment assayed by cell-titer glo (relative luminescent units; $n$ = 4 biological replicates).

D    Cumulative population doublings of KP and KPΔNF1 cells cultured with 2.0 or 0.5 mM glutamine ($n$ = 4 biological replicates).

E    Two patient-derived *KRAS;NF1*-mutant LUAD cell lines (PDKN1, PDKN2), as well as the control patient-derived *KRAS*-mutant/*NF1*-WT LUAD cell line (PDK), were passaged for 14 population doublings prior to collection. The heatmap displays the most significantly upregulated genes in the PDKN1 and PDKN2 cell lines (relative to the PDK control), with the degree of absolute fold-change upregulation depicted by color as indicated in the legend.

F, G    Quantification of the expression of the six key altered genes in cultured PDKN1 and PDKN2 cells (passaged for 14 population doublings) vs. their respective original patient tumor samples using qPCR ($n$ = 4 biological replicates).

H    Western blotting analysis of Psat1 expression in KP and KPΔNF1 cells infected with sgTom or sgPsat1 following selection. GAPDH was used as a loading control.

I    Cumulative population doublings of KP and KPΔNF1 cells after transduction with sgTom- or sgPsat1-containing vectors ($n$ = 4 biological replicates).

J–L    Cumulative population doublings of patient-derived *KRAS*-mutant LUAD cell lines that are either (J, K) *NF1*-mutant (PDKN1 and PDKN2) or (L) *NF1*-WT (PDK) after selection with sgTom- or sgPsat1-containing vectors ($n$ = 4 biological replicates).

Data information: *P*-values are reported in Appendix Table S3. In bar charts and line graphs, data presented as means with error bars representing standard deviations (SDs). For boxplots, whiskers indicate the minimum and maximum values, the upper and lower perimeters represent the first and third quartiles, the midline represents the median value, and the x symbol represents the mean.

Source data are available online for this figure.

research on lung cancer cells has revealed significant changes in gene expression as tumor cells are transitioned to culture (Daniel *et al*, 2009), we also compared the mRNA expression of our six key genes in passaged PDKN1 and PDKN2 cells against their mRNA expression in corresponding PDKN1 and PDKN2 tumors, respectively. We found no significant differences in the mRNA expression of our six key genes (Fig 3F and G), suggesting that passaging in culture does not profoundly influence *NF1*-reactive gene expression in NSCLC cells.

Considering that *NF1*-null cells exhibit heightened glutamine dependence (Fig 3A–D, Appendix Fig S4), we selected the glutamine-metabolizing enzyme *Psat1* for further investigation. Targeted downregulation of Psat1 using sgPsat1 (Fig 3H) conferred a significant growth disadvantage (i.e., up to eightfold decrease in cumulative population doublings measured over 6 days) in murine KPΔNF1 cells (Fig 3I) as well as patient-derived *NF1*-mutant PDKN1 and PDKN2 cells (Fig 3J and K). However, downregulation of Psat1 did not affect patient-derived *NF1*-WT PDK cells (Fig 3L).

## Nf1 loss enhances Kras-mutant sensitivity to glutaminase or Psat1 inhibition

We investigated whether the strong sensitivity of *Kras;Nf1* mutants to glutamine could be therapeutically exploited. The chosen therapeutic strategy was based on the glutamine processing pathway in LUAD cells; wherein, we probed the ability of chemical inhibitors against glutaminase (CB-839) and Psat1 (AOA) to modulate cancer cell growth by restricting the supply of α-KG to the Krebs cycle (Fig 4A). Cellular viability was dose-dependently reduced in KPΔNF1 cells incubated with CB-839 or AOA (Fig 4B and C). Moreover, cellular proliferation was drastically compromised in KPΔNF1 cells incubated with CB-839 or AOA (Fig 4D). In addition, human *KRAS*-mutant lung cancer cell lines harboring *NF1* mutations (i.e., H2030 and H2347) showed marked sensitivity to glutaminase and PSAT1 inhibition, while those with WT *NF1* (i.e., H23, H2009, and H1792) displayed less sensitivity (Fig 4E). This lends support to our aforementioned observations that *Nf1* loss heightens glutamine dependence. In fact, the selectivity of glutamine sensitivity to *NF1* mutations is highlighted when patient-derived *KRAS*-mutant LUAD

cell lines harboring wild-type *NF1* (PDK) or mutant *NF1* (PDKN1 and PDKN2) were treated with CB-839 or AOA. Cell viability was drastically compromised in PDKN1 and PDKN2 cells treated with CB-839 or AOA but not in the similarly treated PDK cells (Appendix Fig S5A and B). Preincubation of KPΔNF1 cultures with glutamate, or the intermediates pyruvate or cell-permeable α-ketoglutarate, rescued the sensitivity to both CB-839 (Appendix Fig S5C–E) and AOA (Appendix Fig S5F–H).

Finally, to determine whether *Nf1*-based glutamine sensitivity requires Fak1 signaling, we utilized two separate scenarios. In the first scenario, the KP clones (with WT *Nf1*) were engineered to harbor gain-of-function *Fak1* cDNA flanked by a doxycycline (DOX)-inducible promoter and HA-tag sequences (KP-ix). PIP2 pretreatment enhanced Fak1 activation, which was further activated by DOX pretreatment; however, DOX pretreatment in the absence of PIP2 did not affect Fak1 activation (Appendix Fig S6A). The expression profiles of the Fak1 target genes (*Psat1*, *Areg*, and *Peg10*) matched that of Fak1 activation (Appendix Fig S6B–D). Just as with inhibitor-treated KPΔNF1 clones, incubating DOX/PIP2-pretreated KP-ix cells with CB-839 or AOA compromised cellular viability (Appendix Fig S6E and F). In the second scenario, KPΔNF1 cells were transduced with WT *Nf1* or vector control cDNAs. As expected, overexpression of WT Nf1 in KPΔNF1 cells downregulates Fak1 (Appendix Fig S6G), downregulates its target genes *Psat1*, *Areg*, and *Peg10* (Appendix Fig S6H–J), retards tumor growth (Appendix Fig S6K), and rescues cytotoxicity induced by CB-839 or AOA (Appendix Fig S6L and M).

## In vivo inhibition of glutaminase or Psat1 effective in controlling Nf1-mutant LUAD

We investigated the effect of *in vivo* inhibition of glutaminase or Psat1 using the $Kras^{LSL-G12D/+}$; $p53^{fl/fl}$ (KP) murine model of *Kras*-driven LUAD (Fig 1). Mice were treated with vehicle control, CB-839, or AOA starting 8 weeks after intratracheal infection of KP mice with lentiviral vectors expressing sgNf1.3 or sgTom as control (Fig 5A). We observed significant decreases in average and total tumor volume in sgNf1.3 KP mice treated with CB-839 or AOA as compared to vehicle-treated sgNf1.3 KP mice (Fig 5B–E). The associated relative response curves for CB-839 (Fig 5F) and AOA (Fig 5G) are provided. At 21 weeks after infection, we also observed

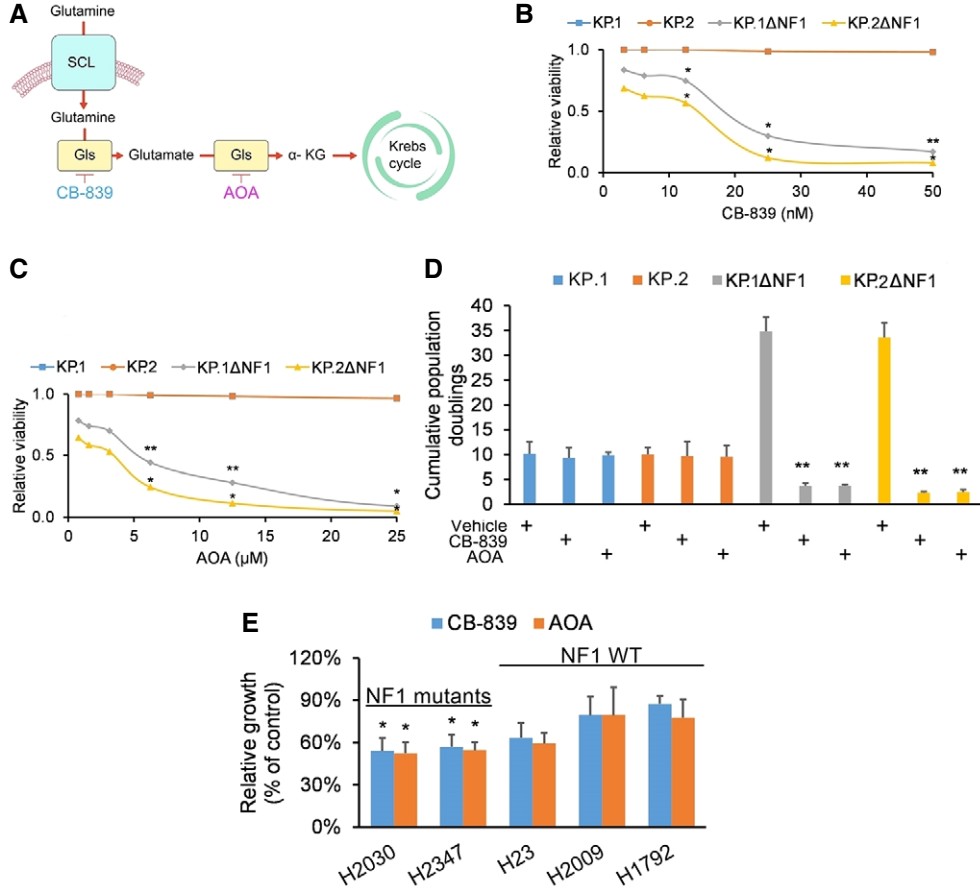

**Figure 4. Nf1-silenced cells display strong sensitivity to glutaminase and aminotransferase inhibition *in vitro*.**

A    Schematic summarizing glutamine processing in LUAD cells: glutamine uptake by SLC transporters, followed by glutamine-to-glutamate conversion by glutaminase (Gls), followed by glutamate-to-α-ketoglutarate (α-KG) conversion by phosphoserine aminotransferase 1 (Psat1). The chemical inhibitors CB-839 and AOA are indicated.

B, C    Relative viability of KP and KPΔNF1 clones after CB-839 treatment (IC$_{50}$: 18 nM) or AOA treatment (IC$_{50}$: 6 μM) for 72 h assayed via CellTiter-Glo (relative luminescence units). All data points are relative to vehicle-treated controls (*n* = 4 biological replicates).

D    Cumulative population doublings of KP and KPΔNF1 cells incubated with vehicle, 18 nM CB-839, or 6 μM AOA after 6 days in culture (*n* = 4 biological replicates).

E    Relative viability of the indicated human *KRAS*-mutant lung cancer cell lines by trypan blue exclusion assay. Cells were incubated with vehicle, 18 nM CB-839, or 6 μM AOA for 72 h (*n* = 4 biological replicates). All data points are normalized to vehicle-treated cell lines.

Data information: *P*-values are reported in Appendix Table S3. In bar charts and line graphs, data presented as means with error bars representing standard deviations (SDs). For boxplots, whiskers indicate the minimum and maximum values, the upper and lower perimeters represent the first and third quartiles, the midline represents the median value, and the x symbol represents the mean.

significant decreases in tumor burden in sgNf1.3 KP mice treated with CB-839 or AOA as compared to vehicle-treated sgNf1.3 KP mice (Fig 5H).

We further investigated the effect of *in vivo* inhibition of glutaminase or Psat1 using subcutaneously and orthotopically transplanted KP tumors. KP and KPΔNF1 cells were subcutaneously transplanted into immunocompromised mice (day 0), and mice were then treated with vehicle control, CB-839, or AOA during days 12–27 (Fig 6A). Tumor growth significantly decreased in inhibitor-treated mice with KPΔNF1 tumors but not inhibitor-treated mice with KP tumors (Fig 6B–D; Appendix Fig S7A and B). A marked reduction in tumor growth was also observed in inhibitor-treated mice with orthotopically transplanted KPΔNF1 tumors but not in inhibitor-treated mice with orthotopically transplanted KP tumors (Fig 6E and F; Appendix Fig S7C and D).

In order to corroborate the *in vitro* observation that *Nf1*-dependent glutamine sensitivity is mediated via Fak1 signaling, KP-ix cells harboring DOX-inducible gain-of-function Fak1 cDNA were subcutaneously transplanted into nude mice followed by treatment with vehicle control, CB-839, or AOA with or without DOX/PIP2 during days 12–24. In the absence of DOX/PIP2, no differences in tumor proliferation were observed between inhibitor-treated mice and vehicle-treated mice, as opposed to an approximately fivefold decrease in tumor volume when using KP-ix (+DOX/PIP2) cells (Fig 6G and H; Appendix Fig S7E and F).

*In vivo* leveraging of glutamine sensitivity as a therapeutic option for human LUAD was carried out using subcutaneous and orthotopic xenograft transplantation of patient-derived *NF1*-mutant (PDKN1 and PDKN2) and *NF1*-WT (PDK) LUAD cell lines into nude mice. Mice were treated with either vehicle, CB-839, or AOA on

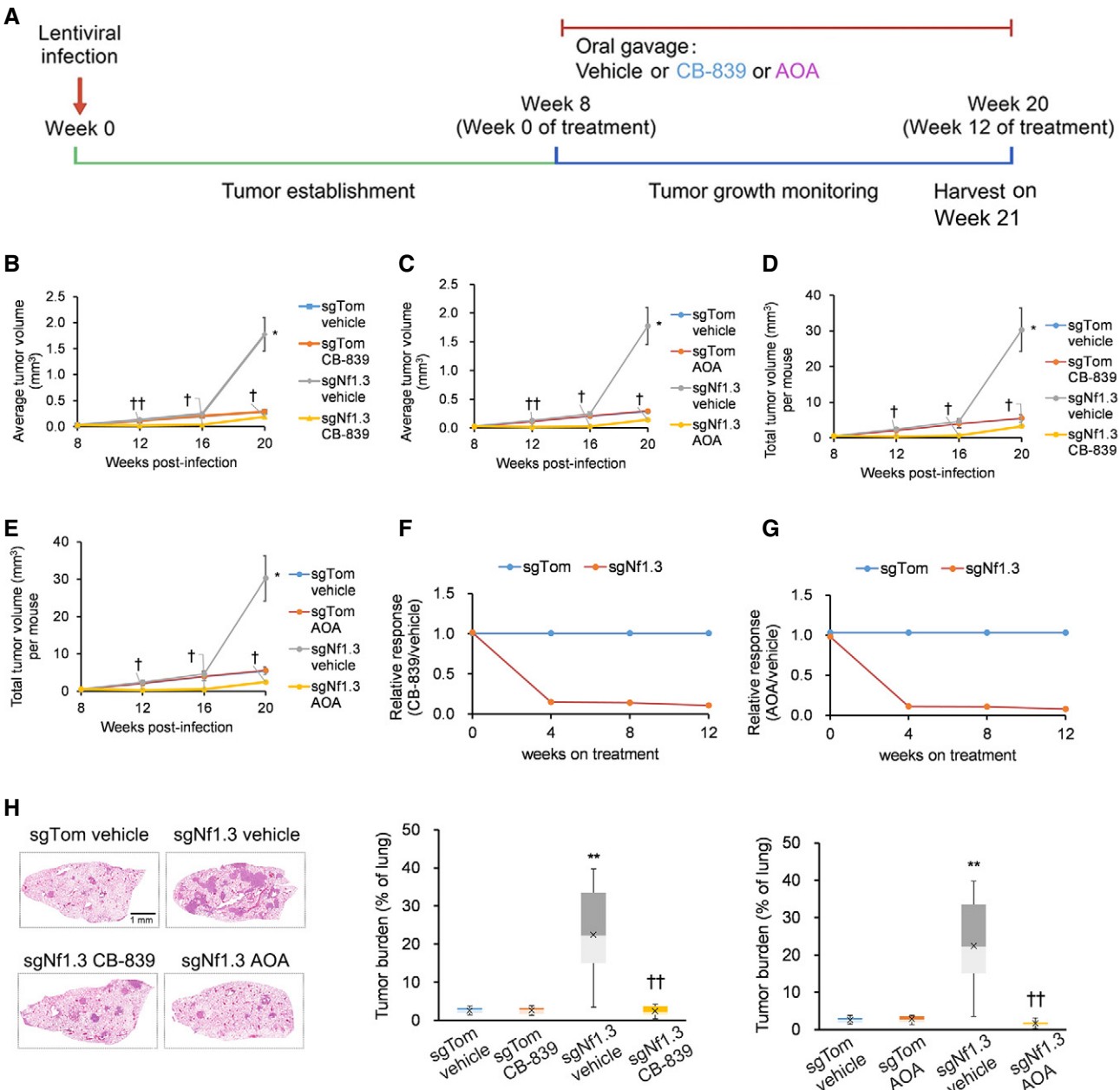

**Figure 5. Nf1/NF1-mutant autochthonous LUAD tumors are sensitive to glutaminase and aminotransferase inhibition *in vivo*.**

A   Schematic describing Kras^LSL-G12D/+; p53^fl/fl (KP) mice intratracheally infected with pSECC lentiviruses containing sgNf1.3 or control sgTom. Mouse tumor burden was tracked by micro-computed tomography (micro-CT) post-infection, and treatment was initiated on week 8 post-infection. Lungs were finally harvested on week 21 post-infection.

B, C   Quantification by micro-CT of mean tumor volumes in (B) CB-839-, (C) AOA-, or vehicle-treated KP mice derived from 50 randomly selected tumors at 8, 12, 16, and 20 weeks post-infection with control sgTom or sgNf1.3. Vehicle-treated KP mice are the same control group for (B) and (C) (n = 21 mice each).

D, E   Quantification by micro-CT of total tumor volume per mouse in (D) CB-839-, (E) AOA-, or vehicle-treated KP mice after infection with control sgTom or sgNf1.3 at 8, 12, 16, and 20 weeks post-infection. Vehicle-treated KP mice are the same control group for (D) and (E) (n = 21 mice each).

F, G   Relative responses of KP mice tumors treated with (F) CB-839, (G) AOA, or vehicle starting from week 8 post-infection and measured until week 20 (n = 21 mice each). Relative response = average tumor volume with treatment/average tumor volume with vehicle.

H   Depictions and quantitation of total tumor burden (total tumor area/total lung area) in CB-839-, AOA-, or vehicle-treated KP mice after infection with control sgTom or sgNf1.3 at 20 weeks post-infection. Vehicle-treated KP mice are the same control group in both boxplots (n = 21 mice each).

Data information: P-values are reported in Appendix Table S3. In bar charts and line graphs, data presented as means with error bars representing standard deviations (SDs). For boxplots, whiskers indicate the minimum and maximum values, the upper and lower perimeters represent the first and third quartiles, the midline represents the median value, and the x symbol represents the mean.

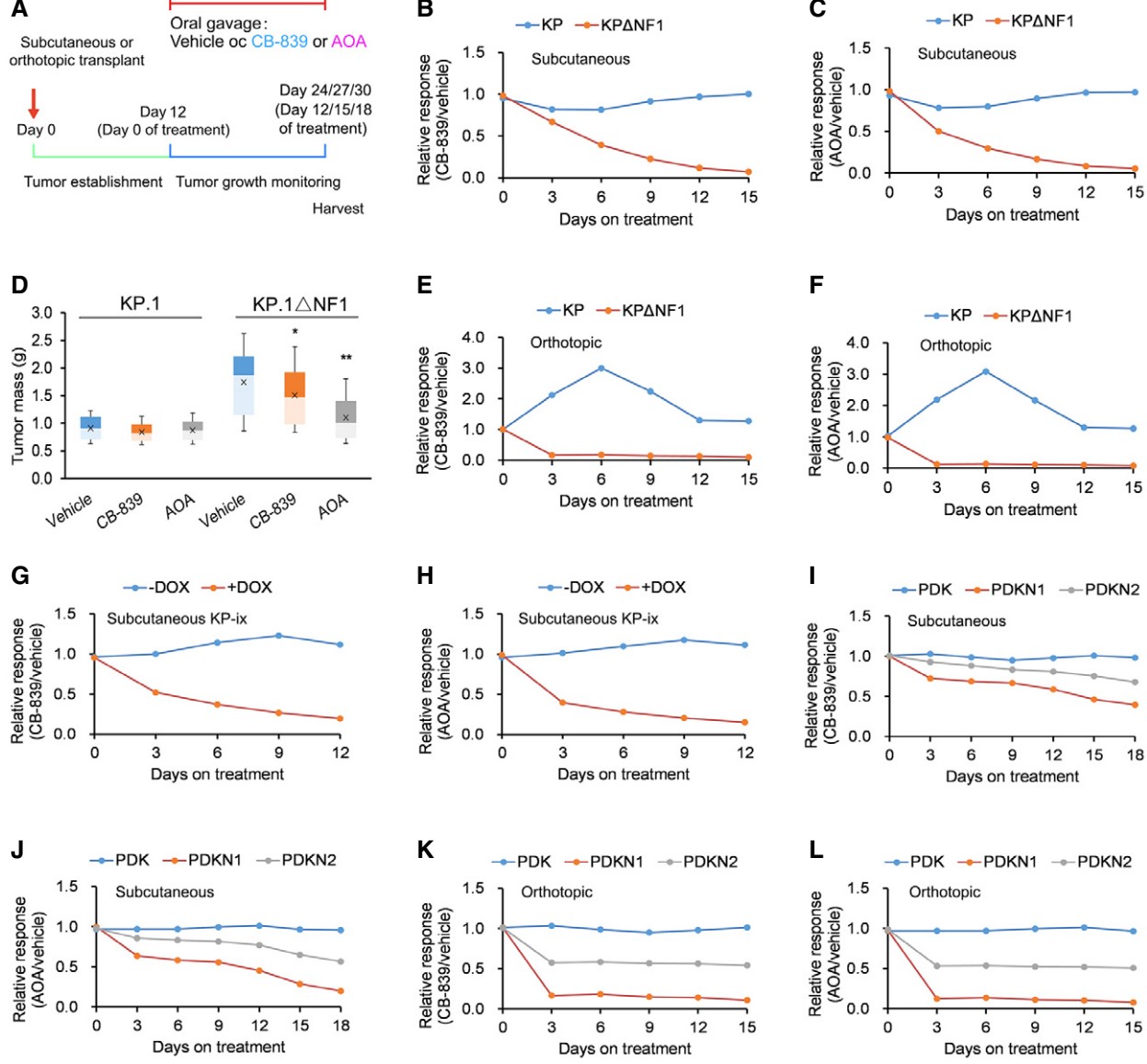

**Figure 6.** *Nf1/NF1*-mutant xenografts are sensitive to glutaminase and aminotransferase inhibition *in vivo*.

A  Schematic describing KP and KPΔNF1 cell transplants into immunocompromised mice.

B–D  Relative responses of subcutaneous KP and KPΔNF1 tumors treated with vehicle, (B) CB-839, or (C) AOA starting from day 12 and measured until day 27 and (D) the final masses of these tumors (*n* = 6 tumors per cell line per treatment). Relative response = average tumor volume with treatment/average tumor volume with vehicle. Full experimental data provided in Appendix Fig S7A and B.

E, F  Relative responses of orthotopic KP and KPΔNF1 tumors treated with vehicle, (E) CB-839, or (F) AOA starting from day 12 and measured until day 27 (*n* = 24 mice per cell line per treatment). Relative response = average tumor volume with treatment/average tumor volume with vehicle. Quantification of tumor growth by photon flux luminescence after orthotopic transplantation with KP or KPΔNF1 cells transduced with a luciferase vector. Full experimental data provided in Appendix Fig S7C and D.

G, H  Relative responses of subcutaneous KP-ix tumors harboring doxycycline (DOX)-inducible gain-of-function (GOF)-Fak1 cDNA treated with vehicle, (G) CB-839, or (H) AOA with or without DOX/PIP2 (*n* = 36 mice per treatment group). Relative response = average tumor volume with treatment/average tumor volume with vehicle. Full experimental data provided in Appendix Fig S7E and F.

I, J  Relative responses of subcutaneous patient-derived *NF1*-mutant (PDKN1 and PDKN2) and *NF1*-WT (PDK) LUAD tumors treated with vehicle, (I) CB-839, or (J) AOA starting from day 12 and measured until day 30 (*n* = 6 tumors per cell line per treatment). Relative response = average tumor volume with treatment/average tumor volume with vehicle. Full experimental data provided in Appendix Fig S7G–I.

K, L  Relative responses of orthotopic patient-derived *NF1*-mutant (PDKN1 and PDKN2) and *NF1*-WT (PDK) LUAD tumors treated with vehicle, (K) CB-839, or (L) AOA starting from day 12 and measured until day 27 (*n* = 12 mice per cell line per treatment). Relative response = average tumor volume with treatment/average tumor volume with vehicle. Quantification of tumor growth by photon flux luminescence after orthotopic transplantation with KP or KPΔNF1 cells transduced with a luciferase vector. Full experimental data provided in Appendix Fig S7J–L.

Data information: *P*-values are reported in Appendix Table S3. For boxplots, whiskers indicate the minimum and maximum values, the upper and lower perimeters represent the first and third quartiles, the midline represents the median value, and the x symbol represents the mean.

days 12–30 (i.e., after the tumor-establishment phase). Subcutaneous tumor growth significantly decreased in inhibitor-treated mice with PDKN1 and PDKN2 tumors but not inhibitor-treated mice with PDK tumors (Fig 6I and J; Appendix Fig S7G–I). Moreover, a marked reduction in tumor growth was also observed in inhibitor-treated mice with orthotopically transplanted PDKN1 and PDKN2 tumors but not in inhibitor-treated mice with orthotopically transplanted PDK tumors (Fig 6K and L; Appendix Fig S7J–L).

## Discussion

More than 95% of *KRAS* gene mutations associated with NSCLC oncogenesis display changes in glycine at position 12 or 13 (Gly12 or Gly13) or glutamine at position 61 (Gln61) in the KRAS protein (Aviel-Ronen *et al*, 2006). These mutations result in a constitutively active KRAS protein, directing cells to proliferate in an uncontrolled manner. Despite having been under investigation for several years, attempts to directly inhibit the biologic activity of the KRAS protein have proved largely unsuccessful (Downward, 2015). As such, characterizing and targeting other functionally relevant molecular aberrations in *KRAS*-mutant LUADs can be used as an alternative approach to managing *KRAS*-mutant LUADs (Skoulidis *et al*, 2015; Romero *et al*, 2017).

In this study, we demonstrated a crucial role of *NF1* loss in *KRAS*-mutant LUAD. Applying CRISPR-Cas9 technology in a murine model, we found that *Nf1* silencing in *Kras*-mutant LUAD is associated with enhanced tumor cell viability, proliferation, endogenous tumor development, and xenograft tumor growth irrespective of p53 mutational status. Accordingly, we observed that *Nf1* overexpression in *Kras*-mutant LUAD is associated with suppressed LUAD cell proliferation and xenograft tumor growth.

A commonplace characteristic of cancer cell metabolism is the ability to quickly acquire and utilize nutrients to satisfy the higher metabolic demands of rapidly proliferating cancer cells (Pavlova & Thompson, 2016). Cancer cells exploit fuels besides glucose, such as glutamine, to assist in core metabolic functions, including energy formation, biomass assimilation, and redox control (Vander Heiden & DeBerardinis, 2017). Specifically, glutamine serves as a key nitrogen-based substrate for nucleotide and amino acid biosynthesis required in rapidly proliferating cancer cells (Altman *et al*, 2016). Here, our initial transcriptomic screen determined that *KRAS;NF1*-mutant LUAD tumors display significant PSAT1 upregulation. Our follow-up experiments revealed that *Nf1* silencing in *Kras*-mutant LUAD upregulates Psat1 in a Fak1-dependent manner. Accordingly, we observed that *Nf1* overexpression in *Kras*-mutant LUAD suppresses Fak1-mediated Psat1 expression. In normal cells, the aminotransferase Psat1 is commonly associated with the reversible catalytic transfer of amino nitrogen between glutamate and phosphoserine in the serine biosynthesis pathway (Possemato *et al*, 2011). However, in breast cancer cells with amplified serine biosynthesis, Psat1 contributes a significant fraction of glutamine-derived carbon flux to α-KG, indicating that Psat1 can play an important role in Krebs cycle anaplerosis of glutamine-derived carbon in cancer cells (Possemato *et al*, 2011). As we observed here, Psat1 upregulation in *Nf1*-mutant LUAD cells heightens glutamine dependence through favoring glutamine-derived carbon flux into the Krebs cycle while reducing glucose-derived carbon flux into the Krebs cycle.

The above findings highlight a particular metabolic vulnerability in *Nf1*-mutant LUAD cells that can be therapeutically exploited. Therefore, we investigated whether increased glutamine addiction could be leveraged as a vulnerability in *Nf1*-mutant LUAD cells. We selected the potent small-molecule glutaminase inhibitor CB-839, which is currently undergoing a phase I clinical trials for solid tumors (NCT02071862, NCT02771626; Choi & Park, 2018) to suppress glutaminase activity because it is rate-limiting in the uptake and utilization of glutamine in cancer cells (Altman *et al*, 2016; Davidson *et al*, 2016). We also employed the small-molecule transaminase inhibitor AOA to suppress Psat1, which lies downstream of glutaminase. We observed that *Nf1* silencing increased the susceptibility of *Kras*-mutant LUAD cells to CB-839 or AOA, while *Nf1* overexpression rescued the cytotoxicity induced by CB-839 or AOA. Moreover, subcutaneous and orthotopic lung implants of *Nf1*-mutant LUAD cells in nude mice exhibited decreased tumor growth when mice were administered CB-839 or AOA, supporting glutaminase and/or Psat1 inhibition as a possible treatment strategy in *Nf1*-mutant LUAD tumors. The foregoing results suggest that LUAD patients can be rationally stratified based on *NF1* mutational status in order to better predict treatment response to PSAT1 inhibition. Although some tumors have shown sensitivity to AOA (Korangath *et al*, 2015; Hao *et al*, 2016), it is a broad-spectrum aminotransferase inhibitor (Altman *et al*, 2016). Therefore, a more specific inhibitor of PSAT1 will be necessary to assess PSAT1's specific role in *NF1*-mutant LUAD.

In addition to the observed cytotoxicity caused by inhibition of glutaminase or Psat1, inhibition of glycolysis by 2-deoxy-d-glucose (2DG) was also potently cytotoxic to *Kras;Nf1*-mutant LUAD cells. It is well-established that glycolytic inhibitors (such as 2DG) selectively induce apoptosis in glucose-dependent cancer cells with overactive *RAS* and *AKT* oncogenes (Kroemer & Pouyssegur, 2008). Specifically, overactive RAS/PI3K/Akt signaling stimulates glycolytic dependence via upregulation of glucose and amino acid flux through the plasma membrane, glucose transporter 1 (GLUT1) expression, GLUT4 translocation to the plasma membrane, and phosphorylation of the glycolytic regulator 6-phosphofructo-2-kinase (Kroemer & Pouyssegur, 2008). As NF1 is a negative regulator of RAS/PI3K/Akt signaling (Shaw & Cantley, 2006), the loss of NF1 function in *Kras;Nf1*-mutant LUAD cells likely results in overactive RAS/PI3K/Akt signaling and heightened dependence on glycolysis. Future research should focus on the glycolytic pathway as a possible metabolic vulnerability in *Kras;Nf1*-mutant LUAD cells.

In conclusion, *Nf1* loss is associated with Fak1 hyperactivation and phosphoserine aminotransferase 1 (Psat1) upregulation in mice. *Nf1* loss also accelerates murine *Kras*-driven LUAD tumorigenesis. The addiction of LUAD tumors harboring *Nf1* mutations to glutamine made them susceptible to glutaminase and Psat1 inhibitors, a strategy that could potentially apply to additional cancers with alterations in the NF1–FAK1 pathway. *NF1* loss has been shown to potentiate resistance to EGFR and BRAF inhibition (Ratner & Miller, 2015), underscoring the significance of this novel treatment approach to combat *KRAS–NF1*-mutant LUAD. More broadly, the findings reported herein detail a CRISPR/Cas9 platform that can reveal metabolic dependencies which can be used to identify novel targets for translational oncologists.

# Materials and Methods

### Ethics statement

This study was approved by the Ethics Committee of the First Affiliated Hospital of Bengbu Medical College (Bengbu, China). All human tissue donors provided written informed consent prior to tissue donation. Experiments with human tissue complied with the principles set forth in the WMA Declaration of Helsinki and the Department of Health and Human Services Belmont Report. The animal procedures were approved by the Ethics Committee of the First Affiliated Hospital of Bengbu Medical College and were performed in accordance with the standards set forth in the Guide for the Care and Use of Laboratory Animals [eighth edition, National Institutes of Health (NIH)].

### Mice

Generation of LSL-$Kras^{G12D}$; $Trp53^{flox}$ (KP) mice was performed as previously described (Jackson $et$ $al$, 2001, 2005). Mice (male and female) maintained on a mixed C57BL/6–129/Sv genetic background were used across all experimental cohorts. Mice aged 6–8 weeks (with the appropriate genotype) were randomized to sgRNA-EFS-Cas92aCre (pSECC)-sgTom (sgTom), pSECC-$sgNf1.1$ (SgNf1.1), pSECC-$sgNf1.2$ (SgNf1.2), or pSECC-$sgNf1.3$ (SgNf1.3) tumor initiation groups. Intratracheal lentiviral infections were performed by a blinded investigator as described (DuPage $et$ $al$, 2009). The sgRNA sequences are detailed in Appendix Table S1. The sample size was based on previous KP mice studies in conjunction with the ARRIVE recommendations on refinement and reduction of animal use in research. All mice were housed under a controlled 12-h/12-h light–dark cycle and with access to food and water $ad$ $libitum$.

### Tumor burden, histological grading, and immunohistochemistry

Mice were sacrificed by carbon dioxide asphyxiation. Lung tissue was fixed overnight by tracheal application of 4% paraformaldehyde, followed by immersion in 70% ethanol, embedding in paraffin, sectioning (4-μm-thickness), and staining with H&E. Postmortem tumor burden was quantified in a blinded fashion. Tumor burden was measured as the percentage of total tumor area relative to total lung area. To calculate total tumor area, individual tumor areas were summed using H&E-stained slides (Nikon 8Oi microscope equipped with a Nikon DS-Ri1 camera and NIS-Elements software, Nikon Corporation, Tokyo, Japan). Total lung area was measured using Panoramic viewer software (3DHISTECH, Budapest, Hungary).

Tumors were graded by a blinded, licensed pathologist using standard histopathological grading techniques (Jackson $et$ $al$, 2005). Immunohistochemical (IHC) staining analyses were blinded such that the experimenter was unaware of the sample genotypes. A DISCOVERY XT instrument (Roche) was employed to perform chromogenic IHC through an automated procedure with the manufacturer's reagents and detection kits. Antibodies to Ki-67 (1:400; Spring Bioscience), pHH3 (Ser10, 1:200; Cell Signaling), and phospho-Fak1 (1:1,000; catalog # sc-374668, Santa Cruz Biotechnology) were used for IHC, followed by HRP detection. Antigen retrieval for phospho-Fak1 was accomplished in Ventana's Cell Conditioner 1 (TBE) while Ki-67 and pHH3 were performed in Cell Conditioner 2 (citrate). Images were captured on a Nikon 8Oi microscope equipped with a Nikon DS-Ri1 camera and processed using NIS-Elements software (Nikon Corporation).

### Human LUAD sample collection and targeted exome capture sequencing

Primary human LUAD tumor samples ($n = 106$), sourced from the Department of Pathology at the First Affiliated Hospital of Bengbu Medical College, were collected along with remote normal lung samples. Tumors were resected and immediately labeled and flash-frozen in liquid nitrogen for later analysis. All protein-coding exons for $KRAS$, $TP53$, $NF1$, and $FAK1$ were sequenced and analyzed as previously described (Romero $et$ $al$, 2017) to identify $KRAS$-mutant/$TP53$-WT/$NF1$-WT/$FAK1$-WT and $KRAS$-mutant/$TP53$-WT/$NF1$-mutant/$FAK1$-WT LUAD tumors.

### Cell culture, treatments, and viability assessment

Parental KP and LKR mouse cell lines were derived from KP and LKR mice, respectively (Meylan $et$ $al$, 2009; Dimitrova $et$ $al$, 2016). We isolated two single-cell clones KP.1 and KP.2 from the KP parental cell line, and we isolated two single-cell clones LKR10 and LKR13 from the LKR parental cell line. One patient-derived $KRAS$-mutant/$TP53$-WT/$NF1$-WT cell line (PDK) and two $KRAS$-mutant/$TP53$-WT/$NF1$-mutant (PDKN1 and PDKN2) cell lines were derived from the human LUAD tumors identified above according to a previously described methodology (Niederst $et$ $al$, 2015). Other human cell lines were purchased from the American Type Culture Collection (ATCC). $Mycoplasma$ tests were routinely performed and were negative for all cultures. Cells were cultured in standard media (DMEM or RPMI) containing FBS (10%) and gentamicin. Cell lines with expression of reverse tetracycline-controlled transactivator (rtTA) or doxycycline-inducible Fak1 constructs were selected and maintained with neomycin (400 μg/ml; Sigma-Aldrich) and hygromycin (600 μg/ml; Sigma-Aldrich), respectively.

Where applicable, cells were treated with (i) aminooxyacetic acid (AOA), a Psat1 inhibitor (Sigma-Aldrich); (ii) CB-839, a glutaminase inhibitor (Calithera Biosciences); (iii) L-buthionine-sulfoximine (BSO; Sigma-Aldrich); (iv) 2-deoxy-D-glucose (2DG; Sigma-Aldrich); (v) dimethyl fumarate (DMF; Sigma-Aldrich); (vi) dimethyl-2-oxoglutarate (DMG, 2 mM; Sigma-Aldrich); (vii) gamma-l-glutamyl-p-nitroanilide (GPNA; Sigma-Aldrich); (viii) glutamate (6 mM; Sigma-Aldrich); (ix) phosphatidylinositol-4,5-bisphosphate (PIP2), a Fak1 activator (Sigma-Aldrich); and (x) pyruvate (2 mM; Gibco).

The viability of cells was assessed using both the CellTiter-Glo™ (Promega) and trypan blue exclusion (Countess II™ Automated Cell Counter Life Technologies) assays. Cellular proliferation was determined via crystal violet staining (prepared in 25:75 methanol:water). For viability assessment of DMF-treated cells, fixation was in paraformaldehyde (4%, 15 min, 4°C), followed by rinsing with ice-cold PBS, and staining with Hoechst dye. Changes in cell densities were quantified on a microplate reader (SpectraMax M5e; Molecular Devices).

## Immunoblotting

Cell lysates were prepared by adding ice-cold RIPA buffer (250 μl; Pierce) containing 1× cOmplete™ Mini inhibitor mixture (Roche). Lysates were mixed on a rotator at 4°C for 30 min, and the protein concentration was quantified using Bio-Rad's DC Protein Assay. SDS–PAGE (4–12% Bis-Tris gradient gel; Bio-Rad) of 50–80 μg total protein per lane followed by immunoblotting was performed according to standard procedures using the following primary antibodies: anti-HA tag (Santa Cruz Biotechnology, sc-57592, 1:1,000), anti-GAPDH (Santa Cruz Biotechnology, sc-25778, 1:500), anti-phospho-Fak1 (Santa Cruz Biotechnology, sc-374668, 1:1,000), anti-total Fak1 (Santa Cruz Biotechnology, sc-1688, 1:1,000), anti-Nf1 (Santa Cruz Biotechnology, sc-74444, 1:1,000), anti-Peg10 (Abcam, ab131194, 1:1,000), anti-Psat1 (Thermo Fisher, PA5-22124, 1:1,000), and anti-p53 (Santa Cruz Biotechnology, sc-393031, 1:1,000).

## qRT–PCR analysis

An RNeasy Plus Mini Kit (Qiagen) was employed to isolate total RNA from cultures. High Capacity cDNA Reverse Transcription Kit (Applied Biosystems) was employed for cDNA synthesis from RNA. Fak1 target genes were quantified by qRT–PCR using a LightCycler 480 II (Roche) using primers outlined in Appendix Table S2.

## Transcriptomic microarray analysis of human LUAD tumor cells

RNA was isolated from cells with the RNeasy Plus Mini Kit (Qiagen). cDNA preparation, probe labeling, hybridization, and array analysis were performed according to the manufacturer's protocols using the Affymetrix Human Genome U133A GeneChip and Affymetrix Microarray software (Affymetrix). Statistical analysis was performed with BioConductor software (http://www.bioconductor.org/), and data preprocessing and normalization were conducted with the Affy package (Affymetrix) as previously described (Golubovskaya *et al*, 2009).

## Assessment of extracellular flux

Analyte-specific extracellular flux measurements were done using Seahorse XF96 Extracellular Flux Analyzer. Quantitative analysis of glucose, lactate, and glutamate turnover was made by comparing fresh media to media collected after 24-h tracing experiments. Cells were counted just prior to quantitative analysis (for normalization) and were assumed to grow exponentially over the culture period.

## Subcutaneous tumor experiments

For the murine LUAD allograft experiments, murine LUAD tumor-derived cells (KP, KPΔNF1, or KP-ix) were randomly allocated and transplanted subcutaneously ($1.0 \times 10^6$ cells) into NOD-SCID-gamma (NSG; NOD.Cg-*Prkdc^{scid}Il2rg^{tm1Wjl}*/SzJ) mice by a blinded investigator. For the subcutaneous patient-derived xenograft experiments, whole tumor specimens (PDKN1, PDKN2, or PDK) were passaged one time through nude mice and then cut into pieces ($2 \times 2$ mm each) that were randomly allocated and subcutaneously transplanted into anesthetized nude mice by a blinded investigator. On day 12 post-implantation, mice harboring subcutaneous tumors were randomly allocated AOA, CB-839, or vehicle as described below. Subcutaneous tumor volumes were calculated as follows: volume ($mm^3$) = $(\pi/6) \times (a^2 \times b)$, wherein $a$ and $b$ are the small and large dimensions, respectively. The sample size was based on previous mice studies in conjunction with the ARRIVE recommendations on refinement and reduction of animal use in research.

## Orthotopic tumor experiments

For the murine LUAD allograft experiments, murine LUAD tumor-derived cells (KP or KPΔNF1) were transduced with a luciferase-expressing vector, randomly allocated, and transplanted orthotopically ($2.5 \times 10^5$ cells) into the left lung of NSG mice under anesthesia by a blinded investigator. For the orthotopic patient-derived xenograft experiments, passaged cells (PDKN1, PDKN2, or PDK) were transduced with a luciferase-expressing vector, randomly allocated, and transplanted orthotopically ($2.5 \times 10^5$ cells) into the left lung of NSG mice under anesthesia by a blinded investigator. The sample size was based on previous mice studies in conjunction with the ARRIVE recommendations on refinement and reduction of animal use in research.

For the orthotopic transplantation procedure (Chatterjee *et al*, 2016), a 1-cm incision was executed over the left scapula. The thoracic musculature was separated to expose the intercostal space. A total of $2.5 \times 10^5$ cells (resuspended in a 10-μl mixture of 1:1 PBS and Matrigel; Zhang *et al*, 2017) were injected via a 29G needle though the intercostal space into the left lung. Incisions were sutured, and topical antibiotics were applied to prevent infection.

On day 12 post-implantation, mice harboring orthotopic tumors were randomized to receive vehicle, CB-839, or AOA as described below. Orthotopic tumor growth was measured by luciferase bioluminescence. Briefly, anesthetized mice were imaged with IVIS Spectrum (PerkinElmer) using sequence luminescence scan mode (5- to 300-s acquisitions). The Living Image 3.2 package was employed to measure tumor average radiance (photon/s/cm$^2$/sr).

## Glutaminase and Psat1 inhibition *in vivo*

After the tumor-establishment phase, mice were treated with CB-839 (200 mg/kg), AOA (25 mg/kg), or vehicle control two times per day. CB-839 was prepared at a concentration of 20 mg/ml so that the total dose volume would be 10 ml/kg. AOA was prepared at a concentration of 2.5 mg/ml so that the total dose volume would be 10 ml/kg. Vehicle control was prepared with 2-hydroxypropyl-β-cyclodextrin (25% w/v) buffered at pH 2 with citrate (10 mM).

## Evaluation of glucose and metabolite tracing by gas chromatography–mass spectrometry (GC-MS)

Cells ($2.0 \times 10^5$ per well) were plated to six-well plates and cultured in 2 ml of RPMI-1640 per well, which was replaced with a 2 ml aliquot of fresh RPMI-1640 media containing [U$^{13}$C]D-glucose (11 mM). Cultures were given 24 h to attain a steady-state labeling of Krebs cycle intermediates. Cultures were then rinsed with ice-cold 1 × PBS, counted (for normalization), and scraped into a 600 μl aliquot of ice-cold methanol (80% v/v) supplemented with norvaline (1.4 μg/ml; Sigma-Aldrich). Cells were homogenized by vortexing (10 min, 4°C) and centrifuged

## The paper explained

### Problem

*KRAS*-based molecular-targeted therapies have been largely unsuccessful due to the difficulties associated with directly inhibiting oncogenic *KRAS*. *KRAS*-mutant LUADs with metabolic susceptibilities can be distinguished based on co-occurring genomic alterations. As such, characterizing and targeting other functionally relevant molecular aberrations in *KRAS*-mutant LUADs can be used as an alternative approach to managing these LUADs.

### Results

This study found that loss of *Nf1* is associated with Fak1 hyperactivation, upregulation of the glutamine-metabolizing enzyme Psat1, and *Kras*-triggered tumorigenesis in a LUAD mouse model. Using analysis of the transcriptome and metabolome, we also demonstrate that tumors with mutation to *Nf1* are reliant upon elevated production of α-ketoglutarate (α-KG) from glutamate via the glutaminase–Psat1 pathway. We also reveal that this metabolic vulnerability can be leveraged as a treatment strategy by pharmacologically inhibiting glutaminase and/or Psat1. Lastly, the work suggests that tumor stratification by co-mutations to *KRAS/NF1* highlights the LAUD patient population expected to benefit from inhibiting PSAT1.

### Impact

The glutamate dependence of LUAD tumors harboring mutations to *Nf1* and their susceptibility to PSAT1 inhibition may potentially be extended to additional cancers with alterations in the NF1–FAK1 pathway. NF1 loss has been shown to potentiate resistance to EGFR and BRAF inhibition, underscoring the benefit of our treatment approach to LUAD tumors with mutations tot KRAS–NF1. More broadly, the report herein details a CRISPR/Cas9 strategy for revealing metabolic dependencies which can be used to identify novel targets for translational oncologists.

(10 min, max speed) to clarify the metabolite-containing supernatant, which was transferred to fresh tubes, dried under nitrogen, and stored frozen until required. Derivatization was done by adding methoxamine (MOX, 16 μl, 60 min, 37°C; Thermo Fisher) and N-tertbutyldimethylchlorosilane (30 min, 60°C; Sigma-Aldrich) followed by analysis with an Agilent 7890A gas chromatograph coupled to an Agilent 5997B mass spectrometer. Helium was used as the carrier gas at a flow rate of 1.2 ml/min. One microliter of sample was injected unto a DB-35 ms column (Agilent Technologies) in a 1:1 split mode at 270°C. The GC oven program following injection was (i) 100°C for 1 min, (ii) gradient rise at 3.5°C/min to 300°C, (iii) gradient rise at 20°C/min to 320°C, and (iv) 320°C for 5 min. For mass analysis, the mass spectrometer was operated under electron impact ionization at 70 eV, with the source and quadrupole held at 230°C and 150°C, respectively. Detection was done in the ion range of 10–650 m/z. Mass isotopomer distributions were determined by integrating appropriate ion fragments for each metabolite as previously described methods (Fernandez *et al*, 1996; Young *et al*, 2008; Lewis *et al*, 2014).

### Statistical analysis

The raw data underlying all charts have been provided in Dataset EV1. Statistical analyses were carried out using methods integrated into GraphPad Prism (version 6.03). Data are reported as means and associated standard deviations (SDs). Shapiro–Wilk testing was used to assess normality. Unpaired two-tailed Student's *t*-testing with *f*-testing was employed to confirm comparable variances between experimental groups. If *f*-testing revealed that variances were significantly different, unpaired two-tailed Student's *t*-testing with Welch's correction was employed. A *P*-value of less than 0.05 was deemed statistically significant.

Expanded View for this article is available online.

## Acknowledgements

This work was supported by the National Natural Science Foundation of China (Grant no. 81772493), the Key Program of Natural Science Research of Higher Education of Anhui Province (Grant no. KJ2017A241), and the Science and Technology Program of Anhui Province (Grant nos. 2017070503B037 and YDZX20183400002554).

## Author contributions

XW and YC designed the study; XW, SM, HL, NW, HW, ZQ, WL, HX, CZ, and YS performed bench experiments; SM, XL, and NW conducted statistical analyses; HX and TW interpreted metabolic data; HW and ZQ collected human tumor samples and characterized and sequenced them; XW and YC wrote the manuscript with comments from all authors.

## Conflict of interest

The authors declare that they have no conflict of interest.

## For more information

Zhang J, Pavlova NN, Thompson CB (2017) Cancer cell metabolism: the essential role of the nonessential amino acid, glutamine. *EMBO J* 36: 1302–1315

Yamaguchi R, Perkins G (2012) Challenges in targeting cancer metabolism for cancer therapy. *EMBO Rep* 13: 1034–1035

Altman BJ, Stine ZE, Dang CV (2016) From Krebs to clinic: glutamine metabolism to cancer therapy. *Nat Rev Cancer* 16: 619–634

Kerr EM, Martins CP (2018) Metabolic rewiring in mutant Kras lung cancer. *FEBS J* 285: 28–41

Kersten K, de Visser KE, van Miltenburg MH, Jonkers J (2017) Genetically engineered mouse models in oncology research and cancer medicine. *EMBO Mol Med* 9: 137–153

Sánchez-Rivera FJ, Jacks T (2015) Applications of the CRISPR–Cas9 system in cancer biology. *Nat Rev Cancer* 15: 387–395

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
