## [Review Process File · EMBO Molecular Medicine]

***Nf1* Loss Promotes *Kras*-Driven Lung Adenocarcinoma and Results in *Psat1*-Mediated Glutamate Dependence**

Xiaojing Wang, Shengping Min, Hongli Liu, Nan Wu, Xincheng Liu, Tao Wang, Wei Li, Yuanbing Shen, Hongtao Wang, Zhongqing Qian, Huanbai Xu, Chengling Zhao and Yuqing Chen

Review timeline:

Submission to The EMBO Journal:	11 September 2018
Editorial Decision:	24 September 2018
Manuscript transferred to EMM:	24 September 2018
Editorial Decision:	15 October 2018
Revision received:	1 February 2019
Editorial Decision:	8 March 2019
Revision received:	25 March 2019
Accepted:	2 April 2019

Editor: Lise Roth

Transaction Report:

Editorial Decision from The EMBO Journal

24 September 2018

Thank you for submitting your manuscript to The EMBO journal and my apologies for the delay in communicating our decision to you. I have now read your study carefully and discussed the work with other members of the editorial team. I am afraid the outcome of these discussions is that we have decided not to pursue publication of this manuscript in The EMBO Journal.

However I have taken the liberty to also discuss your study with my colleague Lise Roth at our sister journal EMBO Molecular Medicine and she would be happy to offer peer review if you were to transfer your manuscript there.

From our side, we appreciate that you find NF1 deletion to increase FAK phosphorylation, cell proliferation and tumor growth in a KRAS-mutated mouse model for LUAD. You go on to show that loss of NF1 or increased expression of FAK leads to a higher glutamine-dependency and therefore to increased sensitivity to small molecule drugs targeting Glis and Psat 1 than seen in standard KRAS mutant cells. However, while you thus provide the first functional evidence for a link between NF1 and KRAS, we are concerned that the two genes had been found to be co-mutated in LUAD. In addition, and as you also mention in the manuscript, we had to notice that NF1 has been independently link to cancer progression and RAS signaling in other contexts.

Consequently, we find that the current manuscript would be a much stronger candidate for publication in EMBO Molecular Medicine, a journal that focuses more on therapeutic and translational impact than on conceptual advance. While I am thus sorry to say that we have decided not to send the manuscript out for peer-review for The EMBO Journal, I would strongly recommend you to transfer the manuscript to EMBO Molecular Medicine following the link provided below. I would like to emphasise that transferring your work to EMBO Molecular Medicine does not involve any reformatting.

The EMBO Journal only subjects those manuscripts to external review that have a high probability of timely publication. Thank you for giving us the opportunity to consider this manuscript. I regret

that we have to disappoint you on this occasion, but hope that you will use this opportunity to transfer your work to EMBO Molecular Medicine.

1st Editorial Decision

15 October 2018

Thank you for the submission of your manuscript to EMBO Molecular Medicine. We have now heard back from the three referees whom we asked to evaluate your manuscript.

As you will see from the reports below, while they all mention the interest and potential clinical relevance of the study, they also raise substantial concerns on your work, which should be convincingly addressed in a major revision of the present manuscript. In particular, drug response in the autochthonous Kras;p53 genetically engineered mouse model should be performed. There is also a need for further strengthening of the data to fully support the conclusions (pFAK quantification, missing controls, data presentation, exhaustive material and methods section, thorough discussion).

Addressing the reviewers concerns in full will be necessary for further considering the manuscript in our journal, and acceptance of the manuscript will entail a second round of review. EMBO Molecular Medicine encourages a single round of revision only and therefore, acceptance or rejection of the manuscript will depend on the completeness of your responses included in the next, final version of the manuscript. For this reason, and to save you from any frustrations in the end, I would strongly advise against returning an incomplete revision.

Please also contact us as soon as possible if similar work is published elsewhere. If other work is published, we may not be able to extend the revision period beyond three months.

I look forward to receiving your revised manuscript.

***** Reviewer's comments *****

Referee #1 (Comments on Novelty/Model System for Author):

More experiments could be done with the primary tumor model used, which is the closest to human LUAD development.

Referee #1 (Remarks for Author):

In their manuscript, the authors use autochthonous mouse models of human LUAD to describe an interesting stronger Fak1 activity in Kras and NF1 mutant LUAD, leading to increased Psat1 expression and increased dependence to glutaminolysis. This effect does not depend on tumor suppressor P53 status. These are original data that may have clinical application in a subset of patients harboring tumors with KRAS and NF1 mutations.

I have a few comments.

Major comments:

1) The authors have identified an NF1-pFak1-Psat1 signaling axis using autochthonous KP tumors, but then they only validate the glutamate dependence and effect of Psat1 or glutaminase inhibitory drugs in cell lines or at best in transplanted tumors. But is this dependence exacerbated by in vitro culture of the cell lines? The authors should test the effects of their drugs (CB-839 and AOA) in the

KP-sgNf1 autochthonous tumors as well, compared to KP-sgTom. Any information obtained from such experiment on this model (effect or no effect on tumor progression) will be valuable to understand how such approaches should be considered in clinical development.

2) What was the extent of Nf1 suppression in sgNf1 autochthonous tumors compared to sgTom?

3) Page 5, the authors say "...confirmed significant p-Fak1 up regulation..." but they refer to a single example of an IHC picture (Fig. 1F). The authors should compare multiple tumors of sgTom and sgNf1.3, quantify the p-Fak1 signal and make an appropriate statistical test to determine if there is any significant difference. In fact, the authors state in the legend that they have monitored n=50 tumors on each arm so the data should be available. These analyses should be clearly from tumors and even from similar grade lesions, contrasting with the example the authors provide where it seems, from the H&E, that the sgTom case is only a hyperplasia. In fact, the H&Es do not seem to match with the IHCs in this Figure. In the legend of this Figure, C refers to D in the Figure, D to E and E to F. "C" is therefore lacking in the legend.

Minor comments:

1) It is surprising that the authors did not represent tumor growth for the KP the same way they represented it for transplanted cells given the fact that autochthonous KP is the best model to see the longitudinal dynamic of tumor progression despite the high heterogeneity of individual tumors within the same animal. Plotting individual tumor growth during the entire tumor development process would be more informative for the reader (instead of total tumor volume as shown in Fig. 1B).

2) Regarding tumor heterogeneity it would be more informative to represent graphs with SD (with individual values would be even better) more than with SEM that does not allow the reader to see the heterogeneity of the samples.

3) Authors described NF1 deficient cells to be more proliferative than control cells and this should be considered for different assays notably those relative to metabolism concerning glucose and glutamine consumption (more cells will consume more glucose). An easy way to rule out such effect is to normalize by the number of cells or quantity proteins just after the experiment.

4) Page 4, the Results section starts by "We next..." as if this was not the beginning of the Results. Throughout the text, there are many instances where words are not separated from one another.

Referee #2 (Remarks for Author):

With the goal of characterizing functionally relevant molecular aberrations in mutant KRAS-driven lung adenocarcinoma (LUAD) as an alternative approach to managing this disease, the authors focused on role of NF1 loss increasing glutaminolysis for the tumor development and homeostasis. Based on the results of several gain-of- and loss-of-function experiments done using in vitro and in vivo models, the authors conclude that NF1 loss hyperactivates FAK1, which in turn upregulates expression of phosphoserine aminotransferase 1 (Psat1), and promotes Kras-driven LUAD mouse model. Furthermore, through transcriptomic and metabolomic analyses, the authors conclude that NF1 loss renders LUAD cells dependent upon increased glutaminolysis. Exploiting these findings, the authors conclude that inhibition of glutaminase or Psat1 genetically or pharmacologically suppress the tumor development and growth. These findings, similar to those of previous studies (Romero and Sayin, 2016, etc.) would provide a rationale for stratification of human patients harboring KRAS/NF1-mutant LUAD tumors for PSAT1 inhibition as a therapeutic option.

This study tests an interesting hypothesis using innovative tools, and its results would help move towards the goal that the authors envision in developing means to clinical application. However, there are several concerns below for which appropriate responses are necessary.

1) Unlike the loss of KEAP1 function, which the authors frequently use as an example of biomarker of synthetic vulnerability for mutant K-RAS, putative loss-of-function alterations of NF1 account for less than 3% (not 6%) of the K-RAS-driven LUADs; In fact, most of NF1 alterations were found

exclusively in oncogene-negative LUADs (Network CGAR, 2014). This mutual exclusivity between K-RAS and NF1 alterations is largely explained by their functional relationship in the same pathway. Nonetheless, the co-occurrence of alterations, no matter how infrequent it is, could justify the goal of study, but the authors need to clarify that.

2) The concept of targeting glutaminolysis in cancer, including K-RAS-driven LUAD, is not new. Additionally, the idea of NF1 loss-driven vulnerability to inhibiting glutaminolysis with CB-839 has recently been shown in soft-tissue tumors (Sheikh, 2017), although it may be new in the context of K-RAS-driven LUAD. Multiple pathways act directly to alter the glutamine metabolism to accommodate the metabolic need of fast proliferating cells, while many oncogenic pathways are indirectly related to the glutaminolysis or merely correlative with. Likewise, the results that the authors use to support the mechanistic link between NF1-FAK1-PAST1 are merely correlative.

3) Recent studies point to the organ/tissue specificity of oncogene-driven metabolic alterations/dependency in cancer including lung adenocarcinoma, in addition to the importance of considering *in vivo* specificity of tumor metabolism. As such, perhaps the most important experiment, in my opinion, would be the genetic and/or pharmacological experiment in the orthotopic transplantation model using of multiple patient-derived NF1-mutant LUAD cell lines. In that regard, the experiment of using subQ model of ***only*** one NF1 mutant patient-derived cell line (shown in Fig. 5I, J) fell short. It is curious why the authors did not perform the orthotopic transplants for this preclinical testing of drug, although they certainly could make one.

4) The quantification of tumor burden and grade in the *in vivo* models ***needs to be*** accompanied with representative images of tumors. Even one panel of H&E image in high magnification is not very telling, given the formation of numerous tumors in one lung.

In addition, some minor changes in the main text and figure will make the manuscript much more read-friendly.

- Throughout the main text, numerous errors in punctuation are noted.
- For most figure panels, it will be great if the authors add a bit more information such as identification of cell, target genes, drug, and incubation time, which should make the panels self-explanatory.
- Line 52, the references are not clear or incomplete and need to be specific.
- Line 66, the first sentence in the Result section begins like " We next investigated...". 'next' means after what?
- Line 81, what does "subcutaneous orthotopic transplants" mean?
- Line 95-97, this text is not clear and does not accurately describe Fig. 2B. No interpretation is given as to why Fak1 level is (already) lower than other cells.
- Line 127, the authors need to provide a succinct rationale for the next experiment before the first sentence.

Referee #3 (Comments on Novelty/Model System for Author):

Drug response studies are primarily performed on the *in vitro* or xenograft models, which may or may not mimic the response in native tumors. Given a previous report that cultured cells exhibit increased glutamine dependency compared to *in vivo* tumors (Davidson et al., Cancer Metabolism, 2016), analysis of therapeutic responses to Psat1 or Glis inhibition in the KP;sgNf1 model is important to strengthen the conclusion of this manuscript.

Referee #3 (Remarks for Author):

By applying the CRISPR-Cas9 technology in a murine model of lung adenocarcinoma (LUAD), the authors demonstrate that loss of the tumor suppressor *Nf1* co-operates with *Kras* mutation to promote tumorigenesis towards higher-grade tumors. In the same model, they claim that loss of *Nf1* hyper-activates *Fak1*. Using murine and human cell culture models, the authors provide mechanistic insights on how the *NF1*-*FAK1*-*Psat1* axis modulates *KRAS* and *NF1* mutant LUAD. They show that *NF1* mutant LUAD cells are dependent on glutamine, and that this vulnerability can be therapeutically exploited by pharmacological inhibition of glutaminase or *Psat1*. This study provides novel insights into the stratification of LUAD based on *KRAS* and *NF1* co-occurring mutations, and suggests that this can be utilized to predict treatment responses to *Psat1* inhibition. This article is timely and provides clinically relevant insights, but it deserves improvements with regards to the validation and characterization of the *in vivo* model. In addition, analysis of treatment responses in more complex *in vivo* settings is recommended to substantiate the major conclusion of this manuscript.

Major comments:

1. The authors claim that loss of *Nf1* increases the tumor burden and promotes progression to high-grade tumors in the *Kras*;p53 model. Representative low-magnification images to depict the tumor burden data of Fig. 1C, 1D, and 1E need to be added to substantiate this finding. Similarly, images with the tumor burden analysis in Fig. 2 and Fig. 5 are required.
2. The authors claim significant upregulation of pFak1 in sg*Nf1.3* KP mice compared to sgTom *Kras*;p53 mice. However, this is not clear from the single image shown in Fig. 1F, and at minimum, quantification of pFak1 IHC data is to be shown, but also qPCR or WB data can substantiate this. Importantly, does the pFak1 positivity depend on tumor grade, and is there intra- and inter-tumor heterogeneity?
3. In Fig. 1, to confirm that loss of *NF1* expression is achieved in KP tumors, authors should demonstrate reduced levels of *Nf1* gene or protein expression, and also demonstrate altered levels of pFak1 and *Psat1* using immunoblotting or qPCR analysis. This will strengthen the finding that loss of *Nf1* connects to this axis in the murine model (see comment 11 also).
4. Data on the three sgRNAs is shown selectively and incompletely in different Figures: In Fig. 1B-E, the data for sg*Nf1.2* is lacking. If there is a difference in the efficiencies among the three sgRNAs, then the motivation for selecting one or two of the sgRNAs need to be added. In addition, the reason for not presenting the data for all *Nf1* guide RNAs need to be included, or data needs to be complemented. E.g., in Fig. 2C-D, knockdown data for all three guide RNAs should be shown. In Fig. 2H-M data for sg*NF1.3* is shown while Fig. 3A-C, and Fig. 4B-D contains data for sg*NF1.1* and sg*NF1.2*; where is it shown that sg*NF1.1* and 1.2 were effective, as earlier it was concluded that 1.3 worked best? Fig. 5B-F lacks information on which of the three guide RNAs was used altogether.
5. The authors describe that overexpression of *NF1* in *KRAS*; *NF1* mutant human cell lines leads to repression of *FAK1* activity as evidenced by downregulation of proposed target genes *PSAT1*, *AREG*, and *PEG10*. Firstly, they should cite a reference where regulation of these genes by *FAK1* has been studied. Secondly, pFAK analysis should be included for the patient-derived LUAD samples shown in Fig. 2E. Thirdly, it needs to be clarified which are the patient-derived LUAD cell lines used in each of the Figures 2F and G, as well as in other places of the document.
6. In Fig. 3E authors show that patient-derived *KRAS*/*NF1*-mutant cells demonstrate high expression of *FAK1* target genes. However, only two patient-derived cultures at passage 14 are used, thus gene expression could be elicited due to cultivation. Can authors quantify the relative expression of altered genes in cultures vs the original patient samples. Also, the legend to Fig. 3E reads that two patient-derived (*KRAS*/*NF1*-mutant LUAD) and one control (*KRAS*-mutant/*NF1*-WT LUAD) lines were used; however, the corresponding text in the Results section on Page 7 refers to KPΔ*NF1* cells.
7. The authors show that in addition to cytotoxicity caused by inhibition of glutaminase or *Psat1*, glycolysis inhibition is also cytotoxic in *KRAS*/*NF1* mutant cells. This suggests that targeting glycolytic pathway constitutes an alternative therapeutic strategy for this subset of lung cancers. This point needs to be addressed in the Discussion.

8. The Methods section on establishment of patient-derived cell lines refers to a study by Niederst, Sequist et al. However, this study appears to primarily focus on cell lines established from EGFR mutant or drug resistant AC or SCLC models. Can authors refer to the ID's of the patient-derived cell lines from Niederst, Sequist et al?

9. In Fig. 5E and F, there appears to be some response at day 12 in the KP group, following 6 days of treatment. However, this is not depicted in the Supplementary Fig. 6C and D, where raw values of the same measurements are plotted.

10. Drug response studies are primarily performed on the in vitro or xenograft models, which may or may not mimic the response in native tumors. Given a previous report that cultured cells exhibit increased glutamine dependency compared to in vivo tumors (Davidson et al., Cancer Metabolism, 2016), analysis of therapeutic responses to Psat1 or Glis inhibition in the KP;sgNf1 model is important to strengthen the conclusion of this manuscript.

11. The following sentence in the Abstract is misleading: 'We discovered that Nf1 loss hyperactivates Fak1, upregulates downstream phosphoserine aminotransferase 1 (Psat1), and promotes Kras-driven LUAD in mice.' The link between loss of Nf1 and upregulation of pFak1 and Psat1 is not well established in the mouse model. Unless the authors provide new data to support this, the sentence needs to be modified to match to the presented data.

12. Interaction with FAK1 is not the main function of NF1. Authors should introduce and discuss their data in the context of known tumor suppressor functions of NF1.

Minor comments:

1. In Fig. 1C-D, on the y-axis remove '%' following the number indicators.
2. Fig. 1D-E are incorrectly labelled in the Legend.
3. In Fig. 2A, authors have used doxorubicin treatment to activate p53. What is the motivation for this? Does this better fit to the Supplementary data? In Fig. 2B, the levels of FAK1 are lower yet pFAK1 is claimed to be activated; protein quantification it needed to derive this conclusion.
4. In Fig. 3B, the unit of the y-axis is missing.
5. In Fig. 4C, authors have used a concentration of AOA that is much above the IC50. Differential drug sensitivity of KP and KPΔNF1 cells is already detected at much lower concentration seen in Fig. 4C. Curve fitting should be done. Similarly, Fig 4B-C five data points are visible for KP cells, whereas six are visible for KPΔNF1 cells.
6. Please provide sequences of the guide RNAs used in the manuscript.
7. In the Legend to Fig. 5, Supplementary Fig. 6 is by mistake named Fig. 7.
8. Please expand on the choice and mechanism of GPNA in the Results section.
9. The Materials and Methods section requires a description of how orthotopic transplantation was performed.

1st Revision - authors' response

1 February 2019

Referee #1 (Remarks for Author):

Major comments:

1) The authors have identified an NF1-pFak1-Psat1 signaling axis using autochthonous KP tumors, but then they only validate the glutamate dependence and effect of Psat1 or glutaminase inhibitory drugs in cell lines or at best in transplanted tumors. But is this dependence exacerbated by in vitro culture of the cell lines? The authors should test the effects of their drugs (CB-839 and AOA) in the KP-sgNf1 autochthonous tumors as well, compared to KP-sgTom. Any information obtained from such experiment on this model (effect or no effect on tumor progression) will be valuable to understand how such approaches should be considered in clinical development.

Response: Thank you for your feedback. We had previously conducted these experiments testing the effects of CB-839 and AOA on KP-sgNf1.3 and KP-sgTom autochthonous tumors, but we had edited them out of the original submission to reduce the manuscript size. As recommended, we have added these experiments back into the revised submission.

2) *What was the extent of Nf1 suppression in sgNf1 autochthonous tumors compared to sgTom?*

Response: We have provided the extent of Nf1 suppression in the sgNf1 autochthonous tumors.

3) *Page 5, the authors say "...confirmed significant p-Fak1 up regulation..." but they refer to a single example of an IHC picture (Fig. 1F). The authors should compare multiple tumors of sgTom and sgNf1.3, quantify the p-Fak1 signal and make an appropriate statistical test to determine if there is any significant difference. In fact, the authors state in the legend that they have monitored n=50 tumors on each arm so the data should be available. These analyses should be clearly from tumors and even from similar grade lesions, contrasting with the example the authors provide where it seems, from the H&E, that the sgTom case is only a hyperplasia. In fact, the H&Es do not seem to match with the IHCs in this Figure. In the legend of this Figure, C refers to D in the Figure, D to E and E to F. "C" is therefore lacking in the legend.*

Response: As recommended, we have quantitatively compared the p-Fak1 IHC signals in sgTom and sgNf1.3 tumors and analyzed this data by tumor grade. We have also corrected the typographical errors in the figure legend.

Minor comments:

1) *It is surprising that the authors did not represent tumor growth for the KP the same way they represented it for transplanted cells given the fact that autochthonous KP is the best model to see the longitudinal dynamic of tumor progression despite the high heterogeneity of individual tumors within the same animal. Plotting individual tumor growth during the entire tumor development process would be more informative for the reader (instead of total tumor volume as shown in Fig. 1B).*

Response: As suggested, we have re-analyzed the raw micro-CT data and added a plot of mean tumor growth starting from Week 8 (Month 2) post-infection (see Figure 1). Week 8 post-infection was chosen as the starting point of longitudinal measurement, since the small size of lung tumors prior to Week 8 made it practically impossible to track single tumors longitudinally by micro-CT before this point.

2) *Regarding tumor heterogeneity, it would be more informative to represent graphs with SD (with individual values would be even better) more than with SEM that does not allow the reader to see the heterogeneity of the samples.*

Response: To better display tumor heterogeneity, we have changed all bar charts to display SDs rather than SEMs.

3) *Authors described NF1 deficient cells to be more proliferative than control cells and this should be considered for different assays notably those relative to metabolism concerning glucose and glutamine consumption (more cells will consume more glucose). An easy way to rule out such effect is to normalize by the number of cells or quantity proteins just after the experiment.*

Response: Yes, these experiments were normalized by the number of cells. We have modified the methods and figure legends to make this clearer to the reader.

4) *Page 4, the Results section starts by "We next..." as if this was not the beginning of the Results. Throughout the text, there are many instances where words are not separated from one another.*

Response: We have had the manuscript edited and proofread by a professional, U.S.-based scientific editing firm. These drafting errors have been corrected throughout the manuscript.

Referee #2 (Remarks for Author):

1) *Unlike the loss of KEAPI function, which the authors frequently use as an example of biomarker of synthetic vulnerability for mutant K-RAS, putative loss-of-function alterations of NF1 account for less than 3% (not 6%) of the K-RAS-driven LUADs; In fact, most of NF1 alterations were found exclusively in oncogene-negative LUADs (Network CGAR, 2014). This mutual exclusivity between K-RAS and NF1 alterations is largely explained by their functional relationship in the same*

pathway. Nonetheless, the co-occurrence of alterations, no matter how infrequent it is, could justify the goal of study, but the authors need to clarify that.

Response: Thank you for pointing out this evidence to us. We have clarified the language in the revised Introduction according to your feedback.

2) *The concept of targeting glutaminolysis in cancer, including K-RAS-driven LUAD, is not new. Additionally, the idea of NF1 loss-driven vulnerability to inhibiting glutaminolysis with CB-839 has recently been shown in soft-tissue tumors (Sheikh, 2017), although it may be new in the context of K-RAS-driven LUAD. Multiple pathways act directly to alter the glutamine metabolism to accommodate the metabolic need of fast proliferating cells, while many oncogenic pathways are indirectly related to the glutaminolysis or merely correlative with. Likewise, the results that the authors use to support the mechanistic link between NF1-FAK1-PAST1 are merely correlative.*

Response: Yes, we concur with your comments. This study is novel with respect to KRAS-driven LUAD and, therefore, provides valuable insights with regard to KRAS–NF1-mutant LUAD tumors. Additionally, we have softened the language in the manuscript to communicate that the mechanistic links between NF1-FAK1-PAST1 are merely correlative.

3) *Recent studies point to the organ/tissue specificity of oncogene-driven metabolic alterations/dependency in cancer including lung adenocarcinoma, in addition to the importance of considering in vivo specificity of tumor metabolism. As such, perhaps the most important experiment, in my opinion, would be the genetic and/or pharmacological experiment in the orthotopic transplantation model using of multiple patient-derived NF1-mutant LUAD cell lines. In that regard, the experiment of using subQ model of ***only*** one NF1 mutant patient-derived cell line (shown in Fig. 5I, J) fell short. It is curious why the authors did not perform the orthotopic transplants for this preclinical testing of drug, although they certainly could make one.*

Response: We had previously conducted subQ and orthotopic transplantation experiments using the two patient-derived NF1-mutant cell lines (PDKN1 and PDKN2) as well as the one patient-derived NF1-WT cell line (PDK), but we had edited some of these experiments out of the original submission to reduce the manuscript size. To address your comment, we have added these experiments back into the revised submission.

4) *The quantification of tumor burden and grade in the in vivo models ***needs to be*** accompanied with representative images of tumors. Even one panel of H&E image in high magnification is not very telling, given the formation of numerous tumors in one lung.*

Response: We have added representative low-magnification images to depict the tumor burden data of Fig. 1. We have also added images associated with the tumor burden analysis in Fig. 2 and Fig. 5.

In addition, some minor changes in the main text and figure will make the manuscript much more reader-friendly. Throughout the main text, numerous errors in punctuation are noted.

Response: We have had the manuscript edited and proofread by a professional, U.S.-based scientific editing firm.

- For most figure panels, it will be great if the authors add a bit more information such as identification of cell, target genes, drug, and incubation time, which should make the panels self-explanatory.

Response: As recommended, we have added more descriptive information to the figure panels where possible.

- Line 52, the references are not clear or incomplete and need to be specific.

Response: We have completely re-written lines 49-53 to address your concerns.

- Line 66, the first sentence in the Result section begins like " We next investigated....". 'next' means after what?

Response: We have removed the word 'next' from this sentence.

- Line 81, what does "subcutaneous orthotopic transplants" mean?

Response: This was a typographical error. The text should have read "...subcutaneous and orthotopic transplants..." We have corrected the language accordingly.

- Line 95-97, this text is not clear and does not accurately describe Fig. 2B. No interpretation is given as to why *Fak1* level is (already) lower than other cells.

Response: To address another reviewer's comment, we have removed the old Fig. 2B and replaced it with an immunoblot focusing on sgNf1.3's effect on p-Fak1 and Fak1 expression. We have also added the protein quantification for p-Fak1 and total Fak1 expression.

- Line 127, the authors need to provide a succinct rationale for the next experiment before the first sentence.

Response: We have provided a succinct rationale as suggested.

Referee #3 (Remarks for Author):

Major comments:

1. The authors claim that loss of *Nf1* increases the tumor burden and promotes progression to high-grade tumors in the *Kras;p53* model. Representative low-magnification images to depict the tumor burden data of Fig. 1C, 1D, and 1E need to be added to substantiate this finding. Similarly, images with the tumor burden analysis in Fig. 2 and Fig. 5 are required.

Response: We have added representative low-magnification images to depict the tumor burden data of Fig. 1. We have also added the charts and low-magnification images associated with the tumor burden analysis in Fig. 2 and Fig. 5.

2. The authors claim significant upregulation of *pFak1* in sgNf1.3 KP mice compared to sgTom *Kras;p53* mice. However, this is not clear from the single image shown in Fig. 1F, and at minimum, quantification of *pFak1* IHC data is to be shown, but also qPCR or WB data can substantiate this. Importantly, does the *pFak1* positivity depend on tumor grade, and is there intra- and inter-tumor heterogeneity?

Response: As recommended, we have quantitatively compared the p-Fak1 IHC signals in sgTom and sgNf1.3 tumors and analyzed this data by tumor grade.

3. In Fig. 1, to confirm that loss of *NF1* expression is achieved in KP tumors, authors should demonstrate reduced levels of *Nf1* gene or protein expression, and also demonstrate altered levels of *pFak1* and *Psat1* using immunoblotting or qPCR analysis. This will strengthen the finding that loss of *Nf1* connects to this axis in the murine model (see comment 11 also).

Response: We have added KP tumor data demonstrating reduced *Nf1* mRNA expression as well as upregulated p-Fak1 protein expression and *Psat1* mRNA expression quantitatively assessed by IHC and qPCR, respectively (see Figure 1).

4. Data on the three sgRNAs is shown selectively and incompletely in different Figures: In Fig. 1B-E, the data for sgNf1.2 is lacking. If there is a difference in the efficiencies among the three sgRNAs, then the motivation for selecting one or two of the sgRNAs need to be added. In addition, the reason for not presenting the data for all *Nf1* guide RNAs need to be included, or data needs to be complemented. E.g., in Fig. 2C-D, knockdown data for all three guide RNAs should be shown.

Response:

--As requested, we have added the sgNf1.2 tumor data to Figure 1.

--We also added the *Nf1* knockdown efficiency data for the three guide RNAs to Figure 1; sgNf1.3 displayed the most potent *Nf1* knockdown efficiency among the three sgRNAs (see Figure 1). Therefore, we chose to use sgNf1.3 as the *Nf1*-silencing sgRNA for all further experimentation. We

have added this rationale to the revised Results section so the reader better understands our reasoning.

-- With regard to your last request and the comments from other reviewers, we have completely revised the immunoblots in Figure 2. Having established that sgNf1.3 is the most efficacious sgRNA in Figure 1, we have removed the old Fig. 2B and replaced it with new immunoblots focusing on sgNf1.3's effect on Nf1, p-Fak1, and Fak1 protein expression in all the tested cell lines in Figure 2. We have also added the quantification for p-Fak1 and total Fak1 protein expression.

In Fig. 2H-M data for sgNF1.3 is shown while Fig. 3A-C, and Fig. 4B-D contains data for sgNF1.1 and sgNF1.2; where is it shown that sgNF1.1 and 1.2 were effective, as earlier it was concluded that 1.3 worked best? Fig. 5B-F lacks information on which of the three guide RNAs was used altogether.

Response: sgNf1.3 was established as the most effective sgRNA in Figure 1, so sgNf1.3 was used for all further experiments in the study. Our nomenclature for the guide RNAs and the corresponding cell lines was unnecessarily confusing. We have renamed the cell lines in a consistent manner to the guide RNAs to avoid confusion. We have also revised the manuscript text to provide information on which guide RNA was used.

5. The authors describe that overexpression of NF1 in KRAS;NF1 mutant human cell lines leads to repression of FAK1 activity as evidenced by downregulation of proposed target genes PSAT1, AREG, and PEG10. Firstly, they should cite a reference where regulation of these genes by FAK1 has been studied.

Response: We have provided the required references as recommended.

Secondly, pFAK analysis should be included for the patient-derived LUAD samples shown in Fig. 2E.

Response: We have provided p-Fak1 and Fak1 immunoblotting analysis for all the cell lines used in Fig. 2.

Thirdly, it needs to be clarified which are the patient-derived LUAD cell lines used in each of the Figures 2F and G, as well as in other places of the document.

Response: As recommended, we have properly defined the patient-derived LUAD cell lines in the revised Methods section and throughout the remainder of the study.

6. In Fig. 3E authors show that patient-derived KRAS/NF1-mutant cells demonstrate high expression of FAK1 target genes. However, only two patient-derived cultures at passage 14 are used, thus gene expression could be elicited due to cultivation. Can authors quantify the relative expression of altered genes in cultures vs the original patient samples?

Response: As recommended, we have quantified the relative expression of the six key altered genes in culture vs. the original patient samples using qPCR and added this information to the revised Figure.

Also, the legend to Fig. 3E reads that two patient-derived (KRAS/NF1-mutant LUAD) and one control (KRAS-mutant/NF1-WT LUAD) lines were used; however, the corresponding text in the Results section on Page 7 refers to KPΔNF1 cells.

Response: We have corrected the text accordingly.

7. The authors show that in addition to cytotoxicity caused by inhibition of glutaminase or Psat1, glycolysis inhibition is also cytotoxic in KRAS/NF1 mutant cells. This suggests that targeting glycolytic pathway constitutes an alternative therapeutic strategy for this subset of lung cancers. This point needs to be addressed in the Discussion.

Response: We have addressed this point in the revised Discussion section.

8. *The Methods section on establishment of patient-derived cell lines refers to a study by Niederst, Sequist et al. However, this study appears to primarily focus on cell lines established from EGFR mutant or drug resistant AC or SCLC models. Can authors refer to the ID's of the patient-derived cell lines from Niederst, Sequist et al?*

Response: There seems to be a misunderstanding of our language. The establishment of our patient-derived cell lines was performed according to the methodology described by Niederst, Sequist et al. We did not use any patient-derived cell lines sourced from Niederst, Sequist et al. or any other research group. We have revised the language to make this clearer to the reader.

9. *In Fig. 5E and F, there appears to be some response at day 12 in the KP group, following 6 days of treatment. However, this is not depicted in the Supplementary Fig. 6C and D, where raw values of the same measurements are plotted.*

Response: For Fig. 5E-F, treatment was started on day 12 and measured until day 27. Therefore, day 12 of treatment in Fig. 5E-F corresponds to day 24 in Supp. Fig. 6C-D. As requested, we have double-checked the raw data and the formulas used to derive the relative response curves to ensure there are no errors in these charts.

10. *Drug response studies are primarily performed on the in vitro or xenograft models, which may or may not mimic the response in native tumors. Given a previous report that cultured cells exhibit increased glutamine dependency compared to in vivo tumors (Davidson et al., Cancer Metabolism, 2016), analysis of therapeutic responses to Psat1 or Glis inhibition in the KP;sgNf1 model is important to strengthen the conclusion of this manuscript.*

Response: As recommended, we have added these experiments testing the effects of CB-839 and AOA on KP-sgNf1 and KP-sgTom autochthonous tumors to the revised work.

11. *The following sentence in the Abstract is misleading: 'We discovered that Nf1 loss hyperactivates Fak1, upregulates downstream phosphoserine aminotransferase 1 (Psat1), and promotes Kras-driven LUAD in mice.' The link between loss of Nf1 and upregulation of pFak1 and Psat1 is not well established in the mouse model. Unless the authors provide new data to support this, the sentence needs to be modified to match to the presented data.*

Response: We have modified the sentence accordingly.

12. *Interaction with FAK1 is not the main function of NF1. Authors should introduce and discuss their data in the context of known tumor suppressor functions of NF1.*

Response: As requested, we have provided NF1's known tumor suppressor functions in the revised Introduction section prior to mentioning FAK1.

Minor comments:

1. *In Fig. 1C-D, on the y-axis remove '%' following the number indicators.*

Response: We have removed the '%' symbols from the y-axis of these figure panels.

2. *Fig. 1D-E are incorrectly labelled in the Legend.*

Response: We have corrected the legend accordingly.

3A. *In Fig. 2A, authors have used doxorubicin treatment to activate p53. What is the motivation for this?*

Response: The first set of experiments demonstrated that *Nf1* loss accelerates *Kras;p53*-mutant LUAD tumorigenesis. However, the effects of *Nf1* loss on *Kras*-mutant;*p53*-WT LUAD tumorigenesis remained unknown. Therefore, we investigated p53's possible involvement in *NF1*-mediated KRAS-LUAD in the *NF1*-null patient-derived LUAD cell lines PDKN1 and PDKN2 and the murine *Kras*-mutant/*p53*-WT/*Nf1*-WT LUAD clones LKR10 and LKR13. Prior to conducting

the experiments, p53 was induced using the DNA intercalator doxorubicin. Doxorubicin has been previously used to induce p53 when conducting experiments using p53-WT tumor cell lines.

Please see:

Romero, R., Sayin, V. I., Davidson, S. M., Bauer, M. R., Singh, S. X., LeBoeuf, S. E., ... & Subbaraj, L. (2017). Keap1 loss promotes Kras-driven lung cancer and results in dependence on glutaminolysis. *Nature medicine*, 23(11), 1362.

Suzuki, H. I., Yamagata, K., Sugimoto, K., Iwamoto, T., Kato, S., & Miyazono, K. (2009). Modulation of microRNA processing by p53. *Nature*, 460(7254), 529.

3B. Does this better fit to the Supplementary data?

Response: We strongly considered your recommendation to move this set of experiments in Figure 2 to the Supplementary Data. However, the other reviewers found Figure 2 to be appropriately positioned in the manuscript, so we have opted to keep it in the main display.

In Fig. 2B, the levels of FAK1 are lower yet pFAK1 is claimed to be activated; protein quantification is needed to derive this conclusion.

Response: Having established that sgNf1.3 is the most efficacious sgRNA in Figure 1, we have removed the old Fig. 2B and replaced it with immunoblots focusing on sgNf1.3's effect on Nf1, p-Fak1, and Fak1 protein expression in all the tested cell lines in Figure 2. We have also added the quantification for p-Fak1 and total Fak1 protein expression.

4. In Fig. 3B, the unit of the y-axis is missing.

Response: We have added the y-axis unit.

5. In Fig. 4C, authors have used a concentration of AOA that is much above the IC50. Differential drug sensitivity of KP and KPΔNF1 cells is already detected at much lower concentration seen in Fig. 4C. Curve fitting should be done. Similarly, Fig 4B-C five data points are visible for KP cells, whereas six are visible for KPΔNF1 cells.

Response: As recommended, we fit the dose-response curves for both CB-839 and AOA and determined their respective IC50 values (18 nM and 6 μM, respectively). We have repeated the *in vitro* experiments using these new IC50-based concentrations. We have also corrected the missing data points in Fig. 4B-C.

6. Please provide sequences of the guide RNAs used in the manuscript.

Response: We have added the guide RNA sequences to the revised Appendix.

7. In the Legend to Fig. 5, Supplementary Fig. 6 is by mistake named Fig. 7.

Response: We have corrected this error.

8. Please expand on the choice and mechanism of GPNA in the Results section.

Response: We have addressed these points in the revised Results section.

9. The Materials and Methods section requires a description of how orthotopic transplantation was performed.

Response: We have added a description of how orthotopic transplantation was performed to the revised Methods section.

Thank you for the submission of your revised manuscript to EMBO Molecular Medicine, and for providing detailed source data. We have now received the enclosed reports from the referees. As

you will see, they are supportive of publication pending minor revisions, and I am pleased to inform you that we will be able to accept your manuscript once the following final editorial amendments will be completed:

1) Referees' comments:

Please address the comments from referees 2 and 3 in the point-by-point response as well as in the manuscript, with the exception of comment 2 from referee 2 (regarding exclusion of the FAK-related data). Indeed, after cross-commenting, the referees agreed that these data were solid and a nice addition to the study, and should thus remain part of the manuscript.

As your study contains lots of data, please be as concise and clear as possible throughout the text.

***** Reviewer's comments *****

Referee #1 (Remarks for Author):

The authors have addressed my comments satisfactorily.

Referee #2 (Remarks for Author):

The authors' response to the reviewer's comments is not the best. It mostly directs the reviewers to the revised text to find changes but did not any marks to indicate them (although the authors said they did). They should have indicated using page number and line, etc.

Overall this revised manuscript has been improved significantly.

Responding to the comments of both reviewers, this revised manuscript includes a significant amount of new data, particularly from animal experiments, which should have been included in the previous version. But some of these new results, especially the images of immunostaining in Figure 1 are suboptimal and not informative. Rather than putting all they have in the figures, some editorial consideration should be there to make concise but robust conclusions.

Regarding my comment 2, however, the authors failed to elaborate mechanistic connections of KRAS-NF1-FAK-PSAT1: the manuscript seems to contain two parts, one for KRAS-NF1-FAK axis and the other for KRAS-NF1-PSAT1. In the end, there was not a good rationale for FAK-PSAT1 connection and FAK seemed lost in discussion. Given the large volume of data, this manuscript would be sufficient if it is rewritten without the parts about FAK.

Referee #3 (Comments on Novelty/Model System for Author):

Comment #2 raises a major concern, only visible following inclusion of the raw data images. We ought to remain vigilant that the KP model data is consistent with data published by the originators of the pSECC-sgtom vector, particularly since this is the reference data to which the Nf1 sg data is compared. It is also a bit concerning that many inconsistencies remain.

Referee #3 (Remarks for Author):

While the authors have addressed most of the comments, but some new issues have arisen or been revealed during the reading of the second version. These are as follows:

1. The following sentence in the Abstract remains misleading: "We discovered that Nf1 loss is associated with Fak1 hyperactivation, phosphoserine aminotransferase 1 (Psat1) upregulation, and Kras-driven LUAD tumorigenesis in mice". However, Kras-driven LUAD tumorigenesis happens even without Nf1 loss. The above sentence needs be restructured, for example "and Nf1 loss also accelerates Kras-driven...". The same conclusion needs to be modified in the Introduction.

2. In Fig. 1B-D, which now includes low-magnification images, shows that there are hardly any tumors in sgTom-treated KP mice at 20 weeks post infection. This result dramatically differs from another study that shows 50-60% tumor burden at 20 weeks pi in KP mice infected with the same virus (Tammela et al., 2017, Fig 2G-H). This is a major issue, and could skew the data to show an effect of Nf1 loss. How do authors explain this discrepancy?
3. While authors now include a quantification of pFAK and pHH3 IHC data, and tumor burden analysis, the methods are not described.
4. The authors need to indicate the tumor grade in Fig. 1H. Also, representative images to depict each tumor grade should be included. Finally, low magnification images of the tumor burden (H&E) and pFAK IHC analyses are still lacking.
5. In the new Fig. 2B-F, all samples are prepared from DOXO-treated cells, making the 'DOXO:' indication above the blot panel redundant. Author may consider writing this as 'PDNK cell line (+ DOXO)'.
6. In several instances (Fig. 3I and Appendix Fig. S1K) cumulative population doublings are shown in the form of line graph, whereas in case of Fig. 4D, it is represented as population doublings using a bar graph (contradicting its legend, which states cumulative population doublings). Also, while in Fig. S1K, the KP-NF1 population doublings are higher than KP at the 6 day time point, in Fig. 4D, the KP cells grow faster. How do authors explain this major inconsistency?
7. While the in vitro study revealed that PIP2 pretreatment is essential for Fak1 activation (Appendix Fig. S6A), such PIP2 pretreatment was not required to expose CB-839 or AOA treatment efficacy in a xenograft established from the same cells (Fig. 6G, H; Appendix Fig. S7E, F). How do authors explain this?

2nd Revision - authors' response

25 March 2019

Referee #2 (Remarks for Author):

The authors' response to the reviewer's comments is not the best. It mostly directs the reviewers to the revised text to find changes but did not any marks to indicate them (although the authors said they did). They should have indicated using page number and line, etc. Overall this revised manuscript has been improved significantly.

Response: Thank you for providing the critical feedback needed to improve our work. Going forward, we will indicate our changes with page numbers and lines.

Responding to the comments of both reviewers, this revised manuscript includes a significant amount of new data, particularly from animal experiments, which should have been included in the previous version. But some of these new results, especially the images of immunostaining in Figure 1 (IHC images in Figure 1H) are suboptimal and not informative. Rather than putting all they have in the figures, some editorial consideration should be there to make concise but robust conclusions.

Response: We have submitted improved immunostaining images for Figure 1H.

Regarding my comment 2, however, the authors failed to elaborate mechanistic connections of KRAS-NF1-FAK-PSAT1: the manuscript seems to contain two parts, one for KRAS-NF1-FAK axis and the other for KRAS-NF1-PSAT1. In the end, there was not a good rationale for FAK-PSAT1 connection and FAK seemed lost in discussion. Given the large volume of data, this manuscript would be sufficient if it is rewritten without the parts about FAK.

Response: Note from the Editor regarding this comment:

After cross-commenting, the referees agreed that these data were solid and a nice addition to the study, and should thus remain part of the manuscript.

Referee #3 (Remarks for Author):

While the authors have addressed most of the comments, but some new issues have arisen or been revealed during the reading of the second version. These are as follows:

1. The following sentence in the Abstract remains misleading: "We discovered that Nf1 loss is associated with Fak1 hyperactivation, phosphoserine aminotransferase 1 (Pstat1) upregulation, and Kras-driven LUAD tumorigenesis in mice". However, Kras-driven LUAD tumorigenesis happens even without Nf1 loss. The above sentence needs to be restructured, for example "and Nf1 loss also accelerates Kras-driven...". The same conclusion needs to be modified in the Introduction.

Response: Thank you for providing the critical feedback needed to improve our work. We have restructured this sentence in the Abstract, Introduction, and Discussion sections.

2. In Fig. 1B-D, which now includes low-magnification images, shows that there are hardly any tumors in sgTom-treated KP mice at 20 weeks post infection. This result dramatically differs from another study that shows 50-60% tumor burden at 20 weeks pi in KP mice infected with the same virus (Tammela et al., 2017, Fig 2G-H). This is a major issue, and could skew the data to show an effect of Nf1 loss. How do authors explain this discrepancy?

Response: We have reviewed the methods and findings in the Tammela et al., 2017 study. We discovered the source of the discrepancy and have provided the supporting references:

- To generate tumors sporadically in the lungs of KP mice, lentiviral particles are used to deliver Cre to the lung.
- In Tammela et al., 2017, tumor burden was measured 20 weeks after infection with 25000 transforming units of sgRNA-EFS-Cas9a2aCre (pSECC)–sgTom lentivirus per mouse.
- However, we used Dupage et al.'s protocol of only 10000 transforming units of pSECC–sgTom lentivirus per mouse. The lower number of Cre-delivering lentiviral particles reduces the number of primary tumors and increases the subjects' average survival time.
- Our tumor burden results are consistent with those of Romero et al.'s work, which also applied Dupage et al.'s protocol.

References

Tammela, T., Sanchez-Rivera, F. J., Cetinbas, N. M., Wu, K., Joshi, N. S., Helenius, K., ... & Gu, X. (2017). A Wnt-producing niche drives proliferative potential and progression in lung adenocarcinoma. *Nature*, 545(7654), 355.

DuPage, M., Dooley, A. L., & Jacks, T. (2009). Conditional mouse lung cancer models using adenoviral or lentiviral delivery of Cre recombinase. *Nature protocols*, 4(7), 1064.

Romero, R., Sayin, V. I., Davidson, S. M., Bauer, M. R., Singh, S. X., LeBoeuf, S. E., ... & Subbaraj, L. (2017). Keap1 loss promotes Kras-driven lung cancer and results in dependence on glutaminolysis. *Nature medicine*, 23(11), 1362.

3. While authors now include a quantification of pFAK and pHH3 IHC data, and tumor burden analysis, the methods are not described.

Response: We have added these methods under their relevant heading in the Methods section.

- Chromogenic IHC was performed as stated under the "Tumor Burden, Histological Grading & Immunohistochemistry" subsection in the Methods section.
- Tumor burden analysis was performed as stated under the "Tumor Burden, Histological Grading & Immunohistochemistry" subsection in the Methods section.

4. The authors need to indicate the tumor grade in Fig. 1H. Also, representative images to depict each tumor grade should be included. Finally, low magnification images of the tumor burden (H&E) and pFAK IHC analyses are still lacking.

Response: As requested, we have indicated the tumor grades for the p-Fak1 IHC images in Figure 1H. We have also added low-magnification H&E and p-Fak1 IHC images to Figure 1H. However, we do not believe images by tumor grade would not add significantly to this figure -- most readers will already be aware of the pathohistological differences by tumor grade, and the differences in p-Fak1 staining by tumor grade (although statistically significant) are not visually striking.

5. In the new Fig. 2B-F, all samples are prepared from DOXO-treated cells, making the 'DOXO:' indication above the blot panel redundant. Author may consider writing this as 'PDNK cell line (+ DOXO)'.

Response: Yes, we agree. We have changed the labeling accordingly.

6. In several instances (Fig. 3I and Appendix Fig. S1K) cumulative population doublings are shown in the form of line graph, whereas in case of Fig. 4D, it is represented as population doublings using a bar graph (contradicting its legend, which states cumulative population doublings).

Response: The Fig. 4D graph's axis labels should read "Cumulative population doublings". We have corrected this in the revised chart.

Also, while in Fig. S1K, the KP-NF1 population doublings are higher than KP at the 6 day time point, in Fig. 4D, the KP cells grow faster. How do authors explain this major inconsistency?

Response: We know the data in Fig. S1K is correct (i.e., KP-NF1 cumulative population doublings are significantly higher than those of KP at Day 6). We have rechecked the experimental raw data, and the data for Fig. 4D was not correctly transferred to the charting program. We have corrected Fig. 4D.

7. While the in vitro study revealed that PIP2 pretreatment is essential for Fak1 activation (Appendix Fig. S6A), such PIP2 pretreatment was not required to expose CB-839 or AOA treatment efficacy in a xenograft established from the same cells (Fig. 6G, H; Appendix Fig. S7E, F). How do authors explain this?

Response: This was a drafting error on our part. In order to activate Fak1, KP-ix cells were treated with DOX and PIP2 (DOX/PIP2) in the same manner in both in vitro and in vivo experiments. This should have been written as "DOX/PIP2" rather than "DOX"; we have corrected this error in the revised manuscript.

Corresponding Author Name: Xiaojing Wang and Yuqing Chen

Journal Submitted to: EMBO Mol Med

Manuscript Number: EMM-2018-09856